

# Exploring 2016-2017 surface ozone pollution over China: source contributions and meteorological influences

Xiao Lu[1,*], Lin Zhang[1,*], Youfan Chen[1], Mi Zhou[1], Bo Zheng[2], Ke Li[3], Yiming Liu[4], Jintai Lin[1], Tzung-May Fu[1], Qiang Zhang[5]

[1]Laboratory for Climate and Ocean-Atmosphere Studies, Department of Atmospheric and Oceanic Sciences, School of Physics, Peking University, Beijing 100871, China

[2]Laboratoire des Sciences du Climat et de l'Environnement, CEA-CNRS-UVSQ, UMR8212, Gif-sur-Yvette, France

[3]John A. Paulson School of Engineering and Applied Sciences, Harvard University, Cambridge, MA 02138, US

[4]Department of Civil and Environmental Engineering, The Hong Kong Polytechnic University, Hong Kong 99907, China

[5]Ministry of Education Key Laboratory for Earth System Modeling, Department of Earth System Science, Tsinghua University, Beijing, China

*Correspondence to*: Lin Zhang (zhanglg@pku.edu.cn) and Xiao Lu (luxiao_atchem@pku.edu.cn)





## Abstract

Severe surface ozone pollution over major Chinese cities has become an emerging air quality concern, raising a new challenge for emission control measures in China. In this study, we explore the source contributions to surface daily maximum 8-h average (MDA8) ozone over China in 2016 and 2017, the two years with the highest surface ozone averaged over Chinese cities in record. We estimate the contributions of anthropogenic, background, and individual natural sources to surface ozone over China, using the GEOS-Chem chemical transport model at $0.25° \times 0.3125°$ horizontal resolution with the most up-to-date Chinese anthropogenic emission inventory. Model results are evaluated with concurrent surface ozone measurements at 169 cities over China and show general good agreement. We find that background ozone (defined as ozone that would be presented in the absence of all Chinese anthropogenic emissions) accounts for 90% (49.4 ppbv) of the national March-April mean surface MDA8 ozone over China and 80% (44.5 ppbv) for May-August. It includes large contributions from natural sources (80% in March-April and 72% in May-August). Among them, biogenic volatile organic compounds (BVOCs) emissions enhance MDA8 ozone by more than 15 ppbv in eastern China during July-August, while lightning $NO_x$ emissions and ozone transport from the stratosphere both lead to ozone enhancements of over 20 ppbv in western China during March-April. Over major Chinese city clusters, domestic anthropogenic sources account for about 30% of the May-August mean surface MDA8 ozone, and reach 39-73 ppbv (38%-69%) for days with simulated MDA8 ozone > 100 ppbv in the Northern China Plain, Fenwei Plain, Yangtze River Delta, and Pearl River Delta city clusters. These high ozone episodes are usually associated with high temperatures, which induce large BVOCs emissions and enhance ozone chemical production. Our results indicate that there would be no days with MDA8 ozone > 80 ppbv in these major Chinese cities in the absence of domestic anthropogenic emissions. We find that the 2017 ozone increases relative to 2016 are largely due to higher background ozone driven by hotter and dryer weather conditions, while changes in domestic anthropogenic emissions alone would have led to ozone decreases in 2017. Meteorological conditions in 2017 favor natural source contributions (particularly soil $NO_x$ and BVOCs ozone enhancements) and ozone chemical production, increase thermal decomposition of peroxyacetyl nitrate (PAN), and further decrease ozone dry deposition velocity. More stringent emission control measures are thus required to offset the adverse effects of unfavorable meteorology such as high temperature on surface ozone air quality.



## 1. Introduction

Ozone near surface is a major air pollutant which harms human health and vegetation growth (Monks et al., 2015). China has become a global hot spot of urban ozone pollution (Wang et al., 2017; Lu et al., 2018a). The present-day (2013-2017) summertime 4[th] highest daily maximum 8-h average (MDA8) ozone levels over eastern China are typically higher than 100 ppbv, inducing significantly larger human health and vegetation damages than those in other industrialized regions (i.e., Japan, Korea, Europe and the

United States) as estimated by different ozone exposure metrics (Fleming et al., 2018; Lu et al., 2018a). In particular, year 2016 and 2017 witnessed the most severe surface ozone pollution in record over most Chinese cities. The summertime surface MDA8 ozone levels in 2016-2017 averaged over the major Chinese cities increased by about 20% compared to 2013-2014 (Lu et al., 2018a), despite that the Chinese Action Plan on Air Pollution Prevention and Control implemented in 2013 has significantly reduced

anthropogenic emissions of nitrogen oxides ($NO_x=NO+NO_2$) and alleviated the winter haze pollution (Cheng et al., 2018; Zheng et al., 2018). Exploring source attribution to surface ozone in China is therefore critical to understand the recent ozone increase and to re-examine the air pollution control strategy.

Surface ozone is mainly produced by sunlight-driven photochemical oxidation of volatile organic

compounds (VOCs) and carbon monoxide (CO) in the presence of $NO_x$. These ozone precursors are emitted intensively in China from anthropogenic sources, including emissions from industry, plant power, residential, and transportation (Li et al., 2017a; Zheng et al., 2018). Significant ozone production driven by intense anthropogenic emissions has been observed and modelled in many urban regions in China (Wang et al., 2006; Ran et al., 2009; Xue et al., 2014; Li et al., 2017b; Tan et al., 2018a). While

anthropogenic contribution can be reduced by emission control measures, background ozone, defined as ozone concentration in the absence of domestic anthropogenic sources, presents a base below which ozone air quality cannot be improved by domestic emission controls. Background ozone includes ozone produced from natural emissions or originated from the stratosphere (together referred as natural background ozone), and ozone produced from foreign anthropogenic emissions via long-range transport,

as described in previous studies in the context of North American background ozone (McDonald-Buller et al., 2011; Zhang et al., 2011; Jaffe et al., 2018). Both natural emissions (e.g., from biosphere or lightning) and transport processes are sensitive to meteorological conditions such as temperature and wind, and can therefore significantly alter the spatiotemporal variability of ozone (Lin et al., 2008; Jacob et al., 2009; Fiore et al., 2012).




Source contribution is not observable and is therefore preferably quantified through modeling studies. Li et al. (2017a) using regional chemical models estimated that anthropogenic emissions from industry, residential, and transportation together contributed up to 80 ppbv ozone (more than half of total ozone) during a severe ozone pollution episode over eastern China in May 2015. For seasonal and nationwide average, however, background sources typically have a larger contribution. Wang et al. (2011) estimated that annually background sources contributed 89% (44.1 ppbv) of mean surface ozone over China in 2006. Ni et al. (2018) also showed a large contribution from background sources to springtime surface ozone in 2008, with more than 40 ppbv over the North China Plain and 70 ppbv over western China. Long-range transport of ozone from Southeast Asia, Europe, and North America can, respectively, enhance regional surface ozone by more than 2 ppbv in China (Ni et al., 2018). Several recent studies also examined the contribution from biogenic VOCs emissions to regional surface ozone (Li et al., 2018a; Mo et al., 2018). No previous studies so far, however, provide a comprehensive view of anthropogenic vs. individual natural source contributions to present-day surface ozone levels in China. Their contributions to recent surface ozone increases over China remain unexplored.

In this study, we explore the sources contributing to surface ozone over China in 2016-2017, using the high resolution (0.25° latitude×0.3125° longitude) GEOS-Chem chemical transport model with the most up-to-date year-specific Chinese anthropogenic emission inventory to interpret the nationwide hourly ozone observation over Chinese cities. We estimate ozone contributions from background, natural, and anthropogenic sources, and further decompose the individual contributions from major natural sources. These include biogenic emissions, soil emissions, lightning emissions, biomass burning emissions, and stratospheric ozone transport. The source attributions and their interactions with meteorology are analyzed both seasonally from a nationwide perspective and at daily scale at individual city for high ozone episodes. We further compare model results for the two years (2016 vs. 2017) to investigate changes in source attributions contributing to recent ozone changes over China.

## 2. Observations and model description

### 2.1 Surface ozone monitoring network

We obtain the nationwide hourly ozone observations from the China National Environmental Monitoring Center (CNEMC) (http://106.37.208.233:20035/). The observational network became operational since 2013, and by 2017 it monitors six surface air pollutants (including ozone and particulate matter with aerodynamic diameter less than or equal to 2.5 μm, PM2.5) in 454 cities (in total of 1597 non-rural sites)





in mainland China (Figure 1). The datasets have been widely used to examine urban air quality issues over China in recent studies (Hu et al., 2017; Li et al., 2017b; Gao et al., 2018; Shen et al., 2018; Li et al.,
2019). Quality controls to remove unreliable hourly observations are applied following our previous work (Lu et al., 2018a).

The Chinese Ministry of Environmental Protection (now the Chinese Ministry of Ecology and Environment) categorized a total of 74 major cities (provincial capital cities and major cities in the North
China Plain (NCP), Yangtze River Delta (YRD), and Pearl River Delta (PRD); three well-developed city clusters in China with severe air pollution) as the State I key cities for air pollution monitoring in 2013. The number of key cities for air pollution monitoring is now expanded to 169 since July 2018 (CNEMC, 2018). The list not only includes more cities in the NCP (now 55 cities), YRD (41 cities), and PRD (9 cities), but also covers regions where air pollution is emerging. As shown in Table 1 and Figure 1, they
are the Fenwei Plain city cluster (FWP, 11 cities), Chengdu-Chongqing city cluster (or Sichuan Basin, SCB, 16 cities), Central Yangtze River city cluster (CYR, 22 cities), and other provincial capital cities in northeastern China (NECH, 4 cities), western China (WCH, 7 cities), and southern China (SCH, 4 cities). We focus the analyses on these 169 cities grouped into the 9 city clusters (Table 1) in this study. Each city contains several monitoring sites, and we average them hourly to represent air quality at the city level.


## 2.2 Model description

We use the GEOS-Chem chemical transport model (CTM) (v11-02rc; http://geos-chem.org) to interpret surface ozone measurements in China. The model is driven by assimilated meteorological data obtained from the Goddard Earth Observing System (GEOS) of NASA Global Modeling and Assimilation Office
(GMAO). The GEOS-FP datasets are available at a native horizontal resolution of 0.25° latitude by 0.3125° longitude, and a temporal resolution of 1h for surface variables and boundary layer height and 3h for others. We use the nested-grid version of GEOS-Chem that has the $0.25° \times 0.3125°$ horizontal resolution over East Asia (70°E-140°E, 15°N-55°N) (Chen et al., 2009; Zhang et al., 2016), with boundary conditions archived from the global simulation at 2° latitude × 2.5° longitude resolution.


The model includes a detailed mechanism of tropospheric $HO_x$-$NO_x$-VOC-ozone-halogen-aerosol chemistry (Wang et al., 1998; Bey et al., 2001; Park et al., 2004; Mao et al., 2013). The chemical kinetics are obtained from the Jet Propulsion Laboratory (JPL) and International Union of Pure and Applied Chemistry (IUPAC) (Sander, et al., 2011; IUPAC, 2013). Photolysis rates are calculated using the Fast-



JX scheme (Bian and Prather, 2002). The linearized ozone parameterization (LINOZ) is used to simulate the stratospheric ozone as described by McLinden et al. (2000). Other species in the stratosphere are calculated in the model based on archived monthly mean production and loss rates provided by the Global Modeling Initiative (GMI) model (Murray et al., 2013). Dry deposition of both gas and aerosols are calculated on-line based on the resistance-in-series algorithm (Wesely 1989; Zhang et al., 2001). The wet

deposition for water-soluble aerosols and gas in GEOS-Chem is described by Liu et al. (2001) and Amos et al. (2012). The non-local scheme for boundary layer mixing process and the relaxed Arakawa–Schubert scheme for cloud convection are described in Lin and McElroy (2010) and Moorthi and Suarez (1992), respectively.

Emissions in GEOS-Chem are processed through the Harvard-NASA Emission Component (HEMCO) (Keller et al., 2014). Global anthropogenic emissions in this study are from the Community Emissions Data System (CEDS, and the latest 2014 condition is used for the model simulation) (Hosely et al., 2018), overwritten by regional emission inventories over the US (National Emission Inventory, NEI), Canada (Canadian Criteria Air Contaminant), Mexico (Kuhns et al., 2005), Europe (European Monitoring and

Evaluation Program, EMEP), Africa (DICE-Africa inventory) (Marais and Wiedinmyer, 2016), and East Asia and South Asia (MIX inventory) (Li et al., 2017a). In particular, we apply the latest Chinese anthropogenic emissions for 2016 and 2017 from the Multi-resolution Emission Inventory for China (MEIC, http://www.meicmodel.org) (Zheng et al., 2018). The MEIC is a bottom-up emission inventory with particular improvements in the accuracy of unit-based power plant emission estimates (Liu et al.,

2015), vehicle emission modelling (Zheng et al., 2014), and the NMVOC speciation method (Li et al., 2014). The bimonthly spatiotemporal distributions of anthropogenic $NO_x$, CO, and non-methane VOCs (NMVOC) emissions averaged for 2016-2017 are shown in Figure S1. Highest emissions exist in the populated city clusters in central eastern China (typically includes NCP, FWP, YRD, CYR, and adjacent regions) with little seasonal variation. Total annual Chinese anthropogenic emissions of $NO_x$, NMVOC,

and CO are, respectively, 22.5, 28.4, 141.9 Tg in 2016, and 22.0, 28.6, 136.2 Tg in 2017 (Zheng et al., 2018). The annual Chinese $NO_x$ and CO emissions both decrease by approximately 20% in 2017 compared to 2013 when the Chinese State Council initiated the Action Plan on Air Pollution Prevention and Control, but NMVOC emissions show a slight increase of 2% (Zheng et al., 2018).

GEOS-Chem has implemented a number of natural emissions, some of which are calculated online in the model. Lightning $NO_x$ emissions are parameterized as a function of cloud top height (Price and Rind,





1992), vertically distributed following Ott et al. (2010), and spatially constrained by climatological observations of lightning flash rates from the Lightning Imaging Sensor (LIS) and the Optical Transient Detector (OTD) satellite instruments (Murray et al., 2012). Following previous studies (Hudman et al., 2007; Zhang et al. 2014), the amount of NO released per flash is 500 moles for the lightning north of 35°N in Eurasia and 23°N in North America, and 260 moles for the rest of the world. Soil $NO_x$ emissions are calculated based on nitrogen (N) availability in soil and edaphic conditions such as soil temperature and moisture as described in Hudman et al. (2010; 2012). Biogenic VOCs (BVOC) emissions are calculated following the Model of Emissions of Gases and Aerosols from Nature (MEGAN version v2.1) algorithm (Guenther et al., 2012). Monthly mean biomass burning emissions for the year 2014 from the Global Fire Emission Database version 4 (GFED4) are used in the simulation (van der Werf et al., 2017). Bimonthly mean total emissions of these natural processes averaged over China are summarized in Table 2. Mixing ratios of methane are prescribed in the model based on spatially interpolated monthly mean surface methane observations from NOAA Global Monitoring Division for 1983-2016, and are extended to 2020 using the linear extrapolation of local 2011-2016 trends (Murray et al., 2016). We find that it leads to ~1% increase in methane concentration over the eastern China in 2017 relative to 2016.

### 2.3 Model configurations

Model configurations are summarized in Table 3. The standard simulation (BASE) includes all anthropogenic and natural emissions as described above. We then conduct sensitivity simulations (1) with all anthropogenic emissions (all emitted pollutants except for methane that is prescribed in all simulations here) turned off over China (noCH) and (2) with all anthropogenic emissions turned off globally (noGLOBE). Ozone concentrations in noCH therefore represent total background ozone in China (hereafter called background ozone), and those in noGLOBE represent natural background ozone (hereafter called natural ozone). The differences in ozone concentration between BASE and noCH denotes the domestic anthropogenic ozone enhancements (CH anthropogenic ozone).

We also conduct several additional sensitivity simulations by turning off individual sources to estimate their ozone enhancements in the presence of all other sources. The ozone contribution from each source can therefore be estimated as the ozone difference between the BASE simulation and each sensitivity simulation. This difference quantifies the source contributions to present-day surface ozone that include the interaction of each specific source with all other sources, rather than the pure impact from the specific source alone (Li et al., 2018a). Four such sensitivity simulations are conducted by turning off (1) BVOC

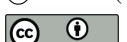



emissions (noBVOC), (2) lightning $NO_x$ emissions (noLIGHT), (3) soil $NO_x$ emissions (noSOIL), and (4)
biomass burning emissions (noBB).

We further quantify the stratospheric contribution to surface ozone using the tagged ozone simulation
(TagO3) (Wang et al., 1998; Zhang et al., 2014), which labels ozone that produced above tropopause from
photolysis of molecular oxygen as stratospheric ozone, and then simulates its transport and chemical loss
in the troposphere (Lu et al., 2019). This tagged stratospheric ozone is calculated based on ozone
production and loss frequency archived from the BASE simulation.

For all simulations (except for the TagO3 simulation which was spun-up for three years), the global
simulation (2° latitude × 2.5° longitude) was first conducted from April 2015 to November 2017. The
results on 1 February 2016 were then interpolated to high resolution (0.25° latitude × 0.3125° longitude)
over the nested domain, and were used to initialize the nested model simulation. Results from the nested
model for March-October in both 2016 and 2017 are analyzed.

## 3. Sources contributing to surface ozone pollution in China
### 3.1 Model evaluation
Figure 2 compares the spatial distributions of observed and simulated bimonthly mean surface MDA8
ozone concentrations at the 169 cities (Fig. 1) averaged for the two years (2016-2017). Figures S2-S3
further compare daily MDA8 values at the nine city clusters and at individual city. We use MDA8
throughout the analysis as it is the form of ozone air quality standard in China and also an important
metric of human health exposure (Turner et al., 2016).

The model reproduces the spatial distribution of observed surface MDA8 ozone in the warm season (May-
August) with a high spatial correlation coefficient (*r*) of 0.81-0.82 and a relatively small positive mean
bias of 4.5-7.4 ppbv (7-13%) at the 169 cities. In particular, the model captures the ozone hot spots over
eastern China (including the NCP, YRD, FWP, and CYR city clusters) in May-June when both
observations and model results show MDA8 ozone > 60 ppbv at a number of cities (Fig. 2). In spring
(March-April) and autumn (September-October), model results have larger positive biases (5.3-9.2 ppbv,
12-20%), while still capture the higher surface ozone in eastern China (*r*=0.32-0.46 for all 169 cities).
The simulated daily MDA8 values are in good agreement with the observed values with *r* ranging from
0.53 to 0.75 at the nine city clusters (Fig. S2). The model captures 57% of the days with observed MDA8 >

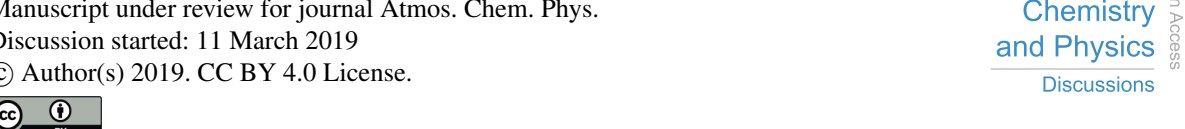

75 ppbv (approximately the grade II Chinese national air quality standard for protection of residential areas) for all nine cities clusters (and more than 60% for NCP, YRD, and FWP cities). The model also reproduces the seasonal variation of surface ozone over mainland China, i.e., the peak in summertime and lower values in other months (Fig. 2). The seasonality of surface ozone typically reflects the dominant role of temperature and solar radiation in enhancing natural emissions of ozone precursors and accelerating photochemical ozone production under high precursor concentrations, with modulations by the arrival of the summer monsoon (Ding et al., 2013; Li et al., 2018b; Lu et al., 2018a).

Surface ozone levels at Chinese urban sites typically show large day-to-day variations driven by intense regional anthropogenic emissions, rapid chemical conversion, and shifts of meteorological patterns (Wang et al., 2006; Lu et al., 2010; Tan et al., 2018b). We find in Figure S3 that observed MDA8 ozone concentrations at many cities can shift by more than 60 ppbv in two days. Such variations, however, are challenging for Eulerian chemical models to capture due to numerical diffusion even at 10km grid resolution and representative issues when comparing gridded simulated results to observations at urban sites (Young et al., 2018). Artificial mixings of ozone precursors in model grid may cause higher ozone production efficiency and therefore positive ozone biases (Wild and Prather, 2006; Yu et al., 2016; Young et al, 2018). In addition, models also have limited skills in presenting local emissions and meteorological conditions particularly over complex terrains (e.g. in WCH and SCB cities) (Zhang et al., 2011; Ni et al., 2018). These limitations largely explain the ozone overestimates at the Chinese urban sites and the deficiencies in capturing extreme high ozone values in our model, as well as many other modeling studies using GEOS-Chem at coarser resolutions (e.g. Wang et al., 2011; Ni et al., 2018) or regional models such as WRF/CMAQ (Chen et al., 2018; Liu et al., 2018). We find that by increasing the resolution from 2°×2.5° to 0.25°×0.3125°, model results show significant improvements in capturing the ozone daily variability and summertime high ozone episodes (model vs. observations correlation coefficients increase from 0.41-0.85 to 0.64-0.88 at individual cities), yet are still not sufficient to reduce the high biases in spring and autumn (Fig. S3).

**3.2 Anthropogenic vs. background ozone contributions over China**

Figure 3 explores the contributions of anthropogenic, background, and natural sources to bimonthly mean surface MDA8 ozone concentration in China averaged for 2016-2017. For the national average (averages of the model grids over the terrestrial land of China), we find that background ozone is the dominant source of the present-day surface MDA8 ozone. Background contributions to surface ozone over China



are as high as 90% (49.4 ppbv) in March-April, followed by 84% (43.2 ppbv) in September-October, 84% (49.7 ppbv) in May-June, and 76% (39.2 ppbv) in July-August (Fig. 3b). Natural sources contribute on average 80% (44.1 ppbv) of the surface MDA8 ozone in March-April and 71% (39.6 ppbv) in May-August, respectively (Fig. 3c). Higher background ozone concentrations are distributed in western and northern China where both natural ozone and ozone imported from foreign anthropogenic sources have larger influences than eastern and southern China. The peak background contribution in spring reflects a mixed effect of larger transboundary transport from foreign anthropogenic sources driven by strong westerly winds (Zhang et al., 2008; HTAP, 2010; Ni et al., 2018), and larger transport from the stratosphere as will be discussed in Section 3.3.

In contrast, ozone contributions from domestic anthropogenic sources are much smaller when averaged over China, with a peak contribution of about 24% (12.1 ppbv) in July-August and less than 15% (5-9 ppbv) in other months (Fig. 3d). The largest anthropogenic ozone enhancements are spatially centered in the central eastern China following the hot spots of anthropogenic emissions (Fig. S1) and population density, and seasonally peaking in summer due to active photochemistry at higher temperature and solar radiation conditions (Fig. S4). There are over 30 ppbv ozone enhancements by domestic anthropogenic sources in central eastern China during July-August that are comparable to background ozone there. Figure 3e shows the spatial distribution of foreign anthropogenic ozone enhancements estimated as the difference between the background ozone (Fig. 3b) and natural ozone (Fig. 3c). Larger contributions are found near the national borders and in the springtime. The foreign anthropogenic contributions we estimate here (~5 ppbv for all seasons averaged over China), however, may not reflect their true contribution, since they are derived in the absence of Chinese domestic anthropogenic emissions and thus do not consider possible interactions with domestic emissions. Ni et al. (2018) using tagged ozone method found that transport from Southern Asia, Japan and Korea, Northern America, and European countries together led to slightly higher springtime surface ozone enhancements of 3-9 ppbv in China with a similar spatial distribution to our results.

Wang et al. (2011) estimated the contribution of background and anthropogenic sources to surface ozone over China in 2006 with an earlier version of GEOS-Chem (v8-01-01) at 0.5°× 0.667° resolution. They used daily mean ozone as metric and found that background ozone accounted for about 93% of total ozone in March-May and 80% in June-August. In good agreement with Wang et al. (2011), our model results when converting to daily mean ozone as metric estimate background ozone contributions of 95% in



March-April and 80% in July-August. These are slightly higher than values derived from the MDA8
metric, reflecting higher domestic anthropogenic contributions to surface ozone at higher ozone levels as
will be discussed later.

**3.3 Identifying ozone enhancements from individual natural sources**

We now illustrate in Figure 4 the impacts of individual natural sources on surface MDA8 ozone
concentrations in China, including ozone enhancements from BVOCs emissions, lightning $NO_x$ emissions,
soil $NO_x$ emissions, biomass burning emissions, and ozone transported from the stratosphere. Here we
focus on the source contributions on bimonthly mean averaged over China, and will present the daily
variations at the city level in Section 3.4.

BVOCs emissions enhance surface MDA8 ozone by 5.1 ppbv in July-August and 2.7-4.1 ppbv in other
months averaged over China (Fig. 4a). The enhancements are particularly large over the central eastern
China where BVOCs enhance surface MDA8 ozone by more than 10 ppbv in July-August. These values
include the interactional effects of BVOCs with other sources. Li et al. (2018a) decomposed the
320 interactional and pure contribution, and found that the interactional contributions of BVOCs and
anthropogenic emissions enhanced summertime (August 2011) surface ozone in urban Xi'an by 14.3 ppbv
that are comparable with our results in this city, while the pure BVOCs contributions alone are only about
2.6 ppbv. The seasonal and spatial variations of BVOCs ozone enhancements (Fig. 4a) are mainly driven
by the exponential dependency of BVOCs emissions on temperature. As shown in Figure S4 and S5, high
biogenic isoprene emissions occur over central eastern China in July-August when daily maximum 2-
meter air temperature ($T_{MAX}$) generally exceeds 30℃. Previous studies have shown that estimates of
BVOCs ozone enhancements can be influenced by the model simulation of yields and fates of organic
nitrates ($RONO_2$), a byproduct from the oxidation of isoprene and its carbonyl (Ito et al., 2009; Fu et al.,
2015). Here the GEOS-Chem model assumes 9% yield of $RONO_2$ from the reaction of isoprene peroxy
radicals with NO, following recent laboratory observations (Paulot et al., 2012;Lee et al., 2014).

Lightning $NO_x$ emissions increase bimonthly mean surface MDA8 ozone by 6.5-9.9 ppbv averaged over
China, with the largest contributions (more than 20 ppbv) found over the Tibetan Plateau (Fig. 4b). In
central eastern China, lightning activities also enhance surface MDA8 ozone by about 2-8 ppbv for all
335 seasons. The spatiotemporal patterns of lightning ozone enhancements, however, do not follow the
lightning $NO_x$ emission patterns that typically peak in summer over central eastern China in the middle



troposphere (600-400hPa) driven by strong convection (Figs S5-S6). The large lightning ozone enhancements over the Tibetan Plateau and the Altai Mountains (in western Mongolia) are mainly due to their high elevations (Fig. S6b). Although lightning $NO_x$ emissions are larger in July-August, the shorter ozone lifetime and stronger upward transport over central eastern China in these months can suppress downward mixing of lightning ozone enhancements to the surface (Fig. S4), resulting in their minimum influences there in the period.

Stratosphere-troposphere exchange (STE) is typically active in the boreal spring at northern mid-latitudes associated with synoptic-scale and mesoscale processes, such as tropopause folds, gravity wave breaking, and deep convections (Stohl et al., 2003). We show in Figure 4c that stratospheric ozone contributions at surface are 8.7 ppbv in March-April and 6.7 ppbv in May-June averaged for China. The western China, particularly the Tibetan Plateau, is strongly influenced by deep stratospheric intrusions with more than 20 ppbv surface MDA8 ozone originated from the stratosphere, consistent with the estimate of Xu et al. (2018) using a different CTM. We also find important stratospheric influences (6-10 ppbv) on surface ozone over the NCP and FWP regions in March-June. Stratospheric contributions to surface ozone are at minimum in summer when STE is weakest and upward transport is strongest limiting downward mixing of stratospheric ozone.

Soil $NO_x$ emissions increase national mean surface MDA8 ozone by 1.6 to 3.1 ppbv (Fig. 4d). Figure S5 shows that the model simulates the highest soil $NO_x$ emissions in May-August over central eastern China and northern Asia, consistent with those derived from satellite observations (Vinken et al., 2014). Large emissions in these regions can be explained by the combined effect of high temperature, and frequent pulsing emissions of soil $NO_x$ after rainfall (Hudman et al., 2012). The largest soil ozone enhancements are not co-located with the emission hotspots, but in regions such as the northern China (2-6 ppbv in May-August) where surface ozone production is more sensitive to $NO_x$ as indicated by the modelled $H_2O_2/HNO_3$ values (Fig. S7). The lower bimonthly mean soil $NO_x$ ozone contributions in central eastern China compared to other regions are due to averaging some negative daily values as will be discussed in Section 3.4.

Compared to other natural sources, biomass burning emissions lead to lower ozone enhancements averaged over China (less than 2 ppbv). This is because biomass burning emissions are generally small over China (less than 2% of the total $NO_x$ emission over China) (Wang et al., 2007; Lin, 2012). Larger



influences are found during boreal spring (March-April) in southwestern China than other seasons and regions in China. The enhancements are particularly large in the southern part of Yunnan Province with over 15 ppbv ozone enhancements driven by the intense fires in the Southeast Asia (Fig. S5d). Previous studies have shown that Eulerian chemical models such as GEOS-Chem tend to overestimate ozone enhancements from biomass burning emissions in regions adjacent to the emission sources (i.e. in southwestern China), but underestimate them in the downwind regions (Zhang et al., 2014; Lu et al., 2016). This can reflect a number of limitations including inadequate representation of wildfire emission and chemistry (e.g., missed short-live VOCs, fewer PAN emissions, and improper emission height), as well as the stretched-flow numerical diffusion of narrow plumes in the model even at fine resolutions (Lu et al., 2016; Eastham et al., 2017).

It should be noted that the total contributions of the above individual natural sources (sum of Fig. 4) together account for about half (from 47.4% in September-October to 53.4% in July-August) of natural ozone levels (Fig, 3c). The missing rest can be largely explained by contributions from global methane. Fiore et al. (2008) found that one Tg emissions of methane would lead to an increase of global mean surface ozone by about 15 pptv (~9 ppbv with present-day annual emissions of 600 Tg) and such sensitivity doubled at high $NO_x$ levels. Further analyses are therefore needed to quantify the contributions of global methane levels to background ozone in China.

### 3.4 Sources contributing to surface ozone in city clusters

It is of particular importance to assess sources contributing to surface MDA8 ozone at the major Chinese cities. Figure 5 shows the ozone contributions from background, anthropogenic, and natural sources at nine city clusters (Fig. 1, Table 1) averaged for March-October (Fig. 5a) and for May-August (Fig. 5b) 2016-2017. For city-cluster averages in both time periods, background sources (and dominantly natural sources) have larger contributions than domestic anthropogenic sources on surface ozone. Averaged for March-October, domestic anthropogenic contributions are 23%, 34%, and 27% for NCP, YRD, FWP city clusters, respectively. The smallest domestic anthropogenic contributions are found over northeastern China (NECH, 10%) and western China (WCH, 18%), where anthropogenic emissions are low and natural source influences from the stratosphere, lightning, and soil are relatively high. For the city clusters in central (CYR, SCB) and southern China (PRD, SCH), Chinese anthropogenic contributions account for 35-42% in March-October. In May-August (Fig. 5b), the domestic anthropogenic contribution fractions increase by about 6-9% at all city clusters except for PRD, and reach 31% in the NCP, 43% in the YRD,





32% in the FWP, reflecting increasing importance of domestic anthropogenic contribution to surface ozone in the peak ozone season. The domestic anthropogenic contribution fractions in PRD remain the same for March-October and May-August averages as surface ozone there typically peaks in September-October (Fig. 2).


In Figure 6, we present the source contributions to daily MDA8 surface ozone in 2017 categorized by different surface ozone levels, which further illustrates the higher contributions of domestic anthropogenic sources to higher daily MDA8 ozone values in Chinese cities. For all city clusters, domestic anthropogenic contributions are much smaller (typically less than 15% of the total simulated ozone) when simulated

MDA8 ozone concentrations are lower than 50 ppbv. These lower contributions are more prevalent during non-summer seasons when ozone is dominantly controlled by background ozone, except for PRD and SCH. Figure 6 also shows that domestic anthropogenic emissions can cause ozone decreases over central eastern China and northern China due to the $NO_x$-saturated chemical regime in spring and fall (Fig. S7), leading to the conditions with background ozone higher than simulated total ozone.


When simulated MDA8 ozone levels are higher than 75 ppbv, domestic anthropogenic contributions significantly increase to 28.8±6.8 ppbv (35%) in NCP, 27.8±7.1 ppbv (33%) in FWP, 38.8±10.7 ppbv (46%) in YRD, 37.7±6.7 ppbv (46%) in SCB, and 54.7±11.8 ppbv (61%) in PRD. Further for the extremely high ozone episodes (simulated MDA8 ozone > 100 ppbv), their contributions are as high as

42.0±5.3 ppbv (41%) in NCP, 39.1±4.8 ppbv (38%) in FWP, 61.4±7.7 ppbv (58%) in YRD, 52.4±10.7 ppbv (48%) in SCB, and 73.2±7.0 ppbv (69%) in PRD. We also find in Figure 6 that there would be no days with MDA8 > 80 ppbv if there were no domestic anthropogenic influences at all 169 major Chinese cities. Our results show that background ozone has a dominant role on the monthly mean surface ozone, on top of which domestic anthropogenic contributions are of particular significance in ozone pollution

episodes. Reducing anthropogenic contributions can therefore be effective in alleviating surface ozone pollution. One exception is the WCH region, where background ozone contributions are dominant for all seasons and even lead to high MDA8 ozone concentrations approaching 80 ppbv.

Figures 7-9 further show the time series of observed vs. simulated daily MDA8 ozone and the source

contributions at representative cities from the three top polluted city clusters (Beijing at NCP, Shanghai at YRD, and Xi'an at FWP) in March-October 2017. The model is able to capture the daily variability of MDA8 ozone as well as the ozone peaks at Shanghai and Xi'an cities (r=0.78 and 0.87, respectively).



However, it fails to reproduce the high ozone episodes in Beijing during May-June 2017, although still with a high temporal correlation coefficient ($r$=0.78), reflecting some model limitations as discussed in
Section 3.1.

Analyses of the daily MDA8 ozone concentrations at the three cities indicate that both the background and domestic anthropogenic ozone enhancements are significantly higher during high ozone episodes than clean days, in consistent with Figure 6. This may reflect their common correlations with
meteorological parameters such as temperature. As shown in Figures 7-9, the deseasonalized summertime (May-August) daily $T_{MAX}$ values are strongly correlated with both observed and simulated ozone, with $r$ ($T_{MAX}$ vs. observed ozone) of 0.72 in Beijing, 0.54 in Shanghai, and 0.75 in Xi'an. We find that such positive temperature-ozone correlations ($r > 0.60$) are widespread in the cities over central eastern China in both observations and model results as shown in Figure 10. High summertime temperatures in China
are typically associated with anti-cyclone systems (e.g., the West Pacific subtropical high) which bring clear sky with strong solar radiation and increase air stagnancy (Pu et al., 2017; Zhao et al., 2017). Such weather conditions tend to enhance ozone chemical production, weaken the pollution ventilation, and favor natural emissions of ozone precursors. We find large temperature-driven daily variations of BVOCs ozone enhancements during May-August in all three cities (Figs 7-9). Taking Shanghai (Fig. 8) as an
example, observed and simulated ozone high episodes are usually associated with large BVOCs ozone enhancements of over 20 ppbv. Nevertheless, in the absence of anthropogenic emissions, the total MDA8 ozone would be below 70 ppbv despite high temperature and background ozone in those cities. Implementing emission control measures under high temperature conditions can be effective to avoid ozone exceedance in the Chinese cities.


Other natural processes such as lightning, soil emissions, and stratospheric transport also contribute to MDA8 ozone daily variations and can episodically lead to large ozone enhancements. For example, at Xi'an, a stratospheric intrusion enhanced surface ozone by over 25 ppbv in early May 2017 and pushed the observed MDA8 ozone approaching 80 ppbv (Fig. 9). Surface ozone enhancements by lightning are
episodically higher than 10 ppbv, and show a similar temporal evolution as stratospheric ozone enhancements, as downward air motion delivers both ozone enhancements originated from high altitudes. Ozone enhancements from soil $NO_x$ emissions become notable when there are pulsing soil emissions after rainfall (e.g., ~10 ppbv in middle August at Xi'an city). They can be either positive or negative reflecting the monthly or even daily shifts of ozone chemical production regime as also reported from satellite

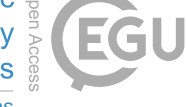

observations (Jin et al., 2015) and several field campaigns in urban China (e.g., Lu et al., 2010). Biomass
burning contributions to surface MDA8 ozone are generally small (less than 2 ppbv, not shown) at the
three cities.

## 4. Factors contributing to the ozone increase from 2016 to 2017

The two-year model results allow us to compare the surface ozone concentrations and their source
contributions between 2016 and 2017. Here we focus on the warm season (May-August) when most cities
in eastern China show high surface ozone levels (Fig. 2). We examine the differences in anthropogenic
and natural emissions, meteorological conditions, and their roles on the surface ozone changes. As shown
in Table 2, domestic anthropogenic emissions of NO and CO in May-August decline by 0.12 Tg (2.5%)

and 1.44 Tg (3.7%) in 2017 compared to 2016, respectively. The decreases of NO and CO emissions are
widespread over eastern China with the largest reductions in Hebei, Shandong, and Jiangsu provinces,
except in Beijing, Shannxi, and Ningxia provinces where both NO and CO emissions show slight
increases (Figs 11a and 11b). On the other hand, anthropogenic NMVOC emissions increase by 0.1 Tg
(1.1%) averaged over China, up to 4% in the NCP (except in Hebei Province) and FWP regions (Fig. 11c).


Figure 12 presents differences in May-August mean meteorological conditions between the two years.
$T_{MAX}$ in the central eastern China averaged for May-August are distinctly higher (> 2K) in 2017 than
2016, together with drier air and more frequent clear sky as indicated by lower specific humidity and
cloud cover fraction in 2017. Such warmer and drier weather conditions in 2017 are favorable for higher

natural emissions of BVOCs and soil $NO_x$. We can see that the model calculates larger biogenic isoprene
(1.36 Tg larger) and soil NO (0.1 Tg larger) emissions in May-August 2017 than those in 2016 over China
(Table 2), mainly over the eastern China (Figs 11d and 11f) following the spatial pattern of temperature
differences. We find that regionally in the eastern China the magnitude of soil NO emissions enhancement
exceeds the decrease of anthropogenic NO emissions, and their total changes over China are also

comparable (Table 2). Lightning $NO_x$ emissions in 2017 are lower than those in 2016 (Fig. 11e).

We show in Figure 13 the observed and simulated May-August mean surface MDA8 ozone differences
between 2016 and 2017. Observed May-August mean surface ozone concentrations show significant
increases in 2017 in the cities over the central eastern China (Fig. 13a). The regional increases are 7.5±6.7

(mean ± standard deviation) ppbv averaged for NCP cities, 12.3±6.7 ppbv for YRD cities, and 12.2±12.2
ppbv for FWP cities. Several cities in PRD show ozone decreases during this period. Although the positive



changes are widely distributed in eastern China, there are large variabilities among cities indicating some local characteristics of ozone changes. Model results with year-specific meteorology and Chinese anthropogenic emissions successfully reproduces the pattern of ozone increases in central eastern China, yet the magnitudes (about 2-6 ppbv) are significantly lower than observations (Fig. 13a). This is likely due to the model limitations as discussed in Section 3.1 and uncertainties in meteorological inputs and anthropogenic emissions. In western China and PRD cities, model results show ozone decreases by about 2 ppbv in 2017 relative to 2016.

Figure 13 also shows the surface ozone changes contributed from different sources as estimated by the model. We find that changes in background ozone alone lead to ozone increases in eastern China particularly in the NCP, FWP, and YRD regions, and decreases in western China (Fig. 13b). The changes of background ozone concentrations are in good agreement with the changes of total ozone for both the spatial pattern and magnitude, and therefore are identified as the main driver of surface ozone changes in 2017 from the 2016 levels. The changes in background ozone are entirely due to changes in meteorological conditions between the two years (except for slight increases in prescribed methane concentrations that are not addressed in the present study). By contrast, changes in domestic anthropogenic emissions generally lead to slight ozone decreases in central eastern China and in the PRD, except for some regions in the Shandong, Hubei, Fujian, and Sichuan provinces (Fig. 13c).

The higher background ozone concentrations in China in May-August 2017 than 2016 can be partly explained by higher contributions from natural sources. Figures 13d and 13e show that the larger emissions of biogenic VOCs and soil $NO_x$ driven by the warmer weather (Figs 11-12) each could have enhanced surface ozone by 1-2 ppbv over central eastern China in 2017. We find that the soil ozone enhancements are more important in driving the surface ozone changes in the northern Asia (e.g., Mongolia and Inner Mongolia), where ozone chemical regimes are highly $NO_x$-sensitive (Fig. S7), while both soil and biogenic emissions enhance surface ozone in central eastern China where ozone productions are sensitive to both $NO_x$ and VOCs during May-August. In addition, the abnormal northwestern winds over Mongolia in 2017 (Fig. 12a) promote the transport of large meteorology-driven soil ozone enhancement to the NCP regions. Although lightning $NO_x$ emissions are smaller in 2017 than 2016, stronger subsidence in the lower troposphere (Fig. 12d) in 2017 allows stronger transport of the lightning produced ozone from the middle/upper troposphere to the surface, contribute slightly higher surface ozone in the NCP and FWP regions by about 1 ppbv (Fig. 13f). This also leads to slightly higher stratospheric



contributions to surface ozone by 0.6 ppbv in 2017.


Furthermore, changes in meteorology can alter tropospheric ozone through modulating the chemical kinetics as discussed in Section 3.4. The warmer and drier weather conditions in May-August 2017 are expected to accelerate ozone production rates, and also favor the thermal decomposition of peroxyacetyl nitrate (PAN, a $NO_x$ reservoir species whose decomposition is strongly temperature-dependent) providing
additional $NO_x$ to produce ozone (Steiner et al., 2010; Doherty et al., 2013). This is evident by the lower background PAN concentrations (model results in noCH that only reflect changes in meteorology) over eastern China simulated in 2017 (Fig. 14a) that follows the spatial pattern of higher temperature (Fig. 12a). Additional influences can come from changes in ozone dry deposition. The model parametrization of dry deposition includes suppression of stomatal uptake of ozone and thus suppression of ozone dry
deposition due to the closure of stomata to protect plants from desiccation at warm air and soil temperature (typically >293K) (Wesely, 1989). We show in Figure 14b that the ozone dry deposition velocities decrease by about 2-10% in May-August 2017 compared to 2016, so that more ozone can remain in the surface layer.

The changes in Chinese anthropogenic ozone as shown in Figure 13c result from the combined effects of changes in domestic anthropogenic emissions and meteorological conditions. While the overall decreases in CH anthropogenic ozone mainly reflect decreasing anthropogenic NO emissions in 2017, the expected ozone reductions due to domestic anthropogenic emission controls may have been largely offset by the higher production rates and lower deposition velocities induced by meteorology as discussed above.
Figure 13c also accounts for the impact of decreasing anthropogenic $PM_{2.5}$ on ozone through affecting heterogeneous chemistry on aerosol surface (i.e. reactive uptake of $HO_2$, $N_2O_5$, $NO_2$, and $NO_3$) and photolysis rates (Lou et al., 2013; Li et al., 2019). A recent study by Li et al. (2019) shows that reducing $PM_{2.5}$ levels by 30-50 $\mu g \ m^{-3}$ (observed changes in 2017 compared to 2013) and associated aerosol optical depths and surface areas in GEOS-Chem leads to about 3-5 ppbv ozone enhancements in eastern China.
We find in Figure S8 that modelled May-August mean anthropogenic $PM_{2.5}$ levels decrease by about 4-10 $\mu g \ m^{-3}$ in central eastern China in 2017 from 2016 levels, which may then lead to about 0.4-1 ppbv ozone enhancements. The above results reflect strong interannual variability of ozone induced by changes in meteorological conditions, and also indicate that more stringent emissions controls on ozone precursors are required in the summertime to attain the air quality standard under unfavorable weather conditions.




## 5. Discussions and Conclusion

In this study, we have estimated the sources contributing to surface MDA8 ozone over China in 2016-
2017, using the state-of-art GEOS-Chem CTM at the 0.25°×0.3125° horizontal resolution with the latest
Chinese anthropogenic emission inventory. These two years show the highest surface ozone
concentrations in record over many major Chinese cities, and ozone increases are observed in most
Chinese cities in 2017 compared to 2016. We show that with the latest anthropogenic emission inventory
and high horizontal resolution, the model well captures the spatial variability of surface ozone in Chinese
cities particularly in the peak ozone season (May-August, r=0.81-0.82) with a small positive mean bias
(6.6 ppbv), although it has some limitations in reproducing the large daily variability in individual cities.

We quantify the contributions from background, natural, and anthropogenic sources and examine their
relative roles in driving the MDA8 ozone changes. We find that at the national scale the model identifies
large contributions from background sources to surface MDA8 ozone concentrations, i.e., 90% (49.4 ppbv)
in March-April and 80% (44.5 ppbv) in May-August averaged over China, including large proportions
from natural sources (80% in March-April and 72% in May-August). Further diagnosing the major natural
sources affecting background ozone, we find that biogenic VOCs emissions alone enhance surface MDA8
ozone by more than 15 ppbv over the central eastern China in July-August mainly driven by high
temperature. Lightning and stratospheric contributions are large (together over 20 ppbv ozone on average
during springtime) in western China due to the high elevation, and can also enhance ozone in the northern
China by over 8 ppbv. Their contributions to ground-level ozone are linked to large-scale vertical transport
and are less important in summer than other seasons. Soil emissions can lead to surface ozone increases
by more than 5 ppbv in the northern China during May-August. Biomass burning emissions are an
important ozone contributor in southwestern China (up to 10 ppbv in Yunnan Province) during March-
April but have low ozone enhancements (less than 2 ppbv) in other regions. In general, we find higher
background ozone concentrations in western and northern China during springtime (March-April) due to
the combined effect of larger natural contributions from the stratosphere and lightning emissions, and
higher ozone imported from foreign anthropogenic sources.

MDA8 ozone contributions from domestic anthropogenic sources are about 19.5% (10.8 ppbv) in May-
August and less than 15% in other months averaged over China. For the NCP, YRD, FWP city clusters,
domestic anthropogenic contributions are, respectively, 23.0%, 34.1%, and 26.5% averaged for March-
October. The contributions increase to 30.7% in the NCP, 43.1% in the YRD, and 32.2% in the FWP in



May-August, reflecting increasing importance of domestic anthropogenic contribution to urban surface ozone in peak ozone seasons. For the extreme high ozone episodes with simulated MDA8 ozone > 100 ppbv, domestic anthropogenic contributions reach 42.0±5.3 ppbv (41%) in NCP, 39.1±4.8 ppbv (38%) in FWP, 61.4±7.7 ppbv (58%) in YRD, and 73.2±7.0 ppbv (69%) in PRD. Our model would predict no days with MDA8 ozone > 80 ppbv in the absence of domestic anthropogenic emissions at all 169 major Chinese cities. We also find that day-to-day ozone evolutions at Beijing, Shanghai, and Xi'an show significant positive correlations with the daily maximum temperature, and such significant correlations are widespread over central eastern China. The high ozone episodes in these cities are typically associated with high temperature weather conditions which cause intensive biogenic VOCs emissions and accelerate ozone production from both background and anthropogenic sources.

We further examine the relative contributions from different sources to surface MDA8 ozone changes in May-August 2017 compared to 2016. The model successfully reproduces the MDA8 ozone increases over central eastern China by 2-5 ppbv, although it underestimates the absolute change compared to the observation. The model attributes the ozone increases to background ozone changes. Higher background ozone concentrations are found in 2017 driven by hotter and dryer weather conditions which accelerate ozone chemical production and increase ozone enhancements from both natural soil $NO_x$ and BVOCs emissions by 1-2 ppbv. Additional impacts of the 2017 meteorology include increasing air subsidence (therefore enhancing lightning and stratospheric ozone contributions), increasing thermal decomposition of PAN, and decreasing ozone dry deposition velocity. The 2017-2016 changes in anthropogenic emissions would have decreased surface MDA8 ozone by about 1 ppbv over central eastern China and 2 ppbv in the PRD regions. Our results indicate that more stringent emission control measures on both $NO_x$ and NMVOCs are needed to reduce present-day surface ozone pollution over China particularly in days and years with unfavorable meteorological conditions.

The focus of this study is to understand the sources contributing to surface ozone over China at present-day (2016 and 2017) emission levels, and it does not address the non-linear changes of source attribution and the effect on surface ozone when anthropogenic emissions levels change in the future. Ni et al. (2018) compared sources contributions estimated from the zero-out methods (used in this study) and the 20% perturbation methods (i.e., reducing emissions of individual sources by 20% in simulations), and found that the absolute values can differ by 2-3 ppbv, whereas the spatial distributions are very similar. Although the 2016-2017 period is warmer than 2013-2015 over most regions in China (Fig. S9), we do not see




continuous increases in temperature over central eastern China from 2013-2017. This implies that
temperature changes alone are not enough to explain the continuously increasing ozone trends in the
Chinese cities over 2013-2017 (Lu et al., 2018a). Other meteorological factors and modes such as the
summer monsoon (Yang et al., 2014; Lu et al., 2018b) and the West Pacific Subtropical High (Zhao et al.,
2017) may also modulate the interannual variability of ozone over China. More modelling studies are still
in need to better understand ozone changes over China in a longer time period and to evaluate emission
control strategies.

**9 figures are shown in the supplement.**

**Author contribution**

L. Zhang and X. Lu designed the study. X. Lu performed model simulations and conducted data analysis
with the assistance of Y.F. Chen, M. Zhou, B. Zheng, K. Li, and Y.M. Liu. J.T. Lin and T.M. Fu assisted
in the interpretation of the results. B. Zheng and Q. Zhang provided the Chinese anthropogenic emissions
inventories. X. Lu and L. Zhang wrote the paper. All authors contributed to the discussion and
improvement of the paper

**Data availability**

The datasets including measurements and model simulations used in this study can be accessed by
contacting the corresponding authors (Lin Zhang, zhanglg@pku.edu.cn; and Xiao Lu,
luxiao_atchem@pku.edu.cn).

**Competing interests**

The authors declare that they have no conflict of interest.

**Acknowledgement**

This work is supported by the National Key Research and Development Program of China
(2017YFC0210102) and the National Natural Science Foundation of China (41475112).



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





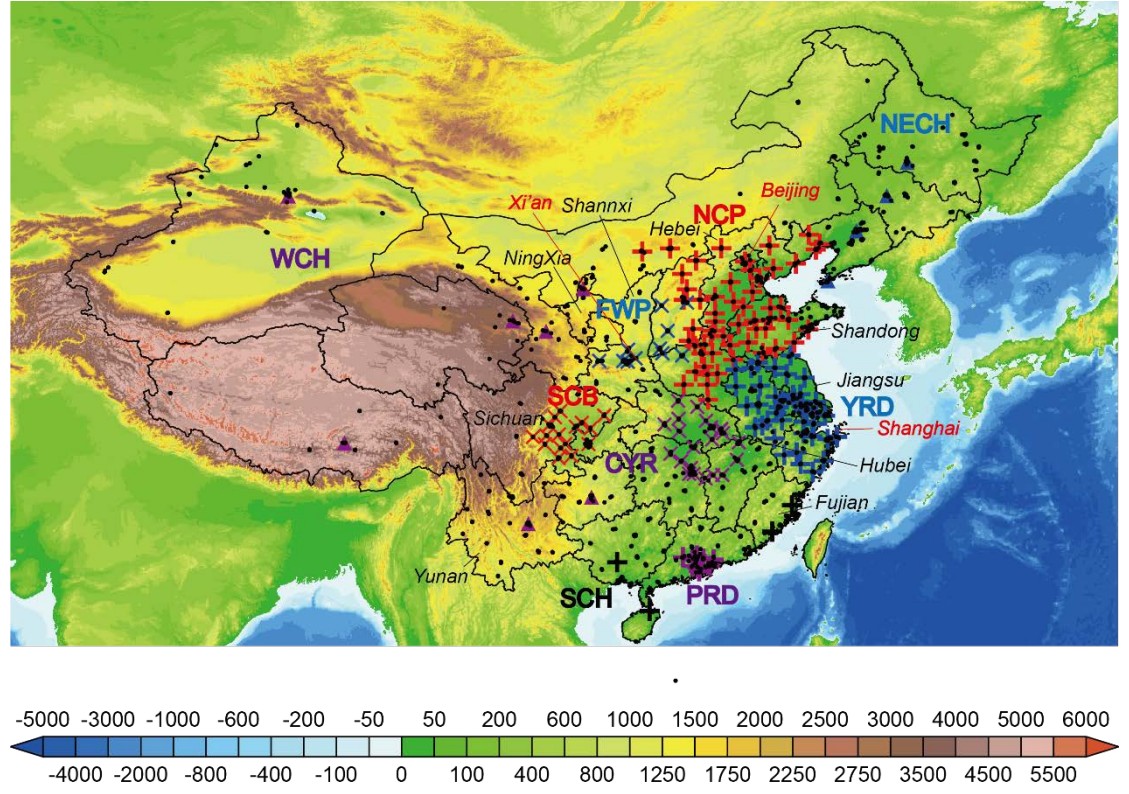

**Figure 1.** Site locations (black dots) of the China National Environmental Monitoring Center (CNEMC)
network. Also shown are the locations of 169 major cities grouped to 9 city clusters (Table 1): the North
China Plain (NCP; red pluses) cluster, the Yangtze River Delta (YRD; blue pluses) cluster, blue crosses
for the Fenwei Plain (FWP; blue crosses) cluster, the Sichuan Basin (SCB; red crosses) cluster, the Central
Yangtze River (CYR; purple crosses) cluster, the Pearl River delta (PRD; purple pluses) cluster, the
northeastern China (NECH; blue triangles) cluster, the western China (WCH; purple triangles) cluster,
and the southern China (SCH; black pluses). The provinces mentioned in the text are labelled in black,
and the three cities analyzed in Figures 7-9 are labeled in red. The underlying figure shows terrain
elevation (m).



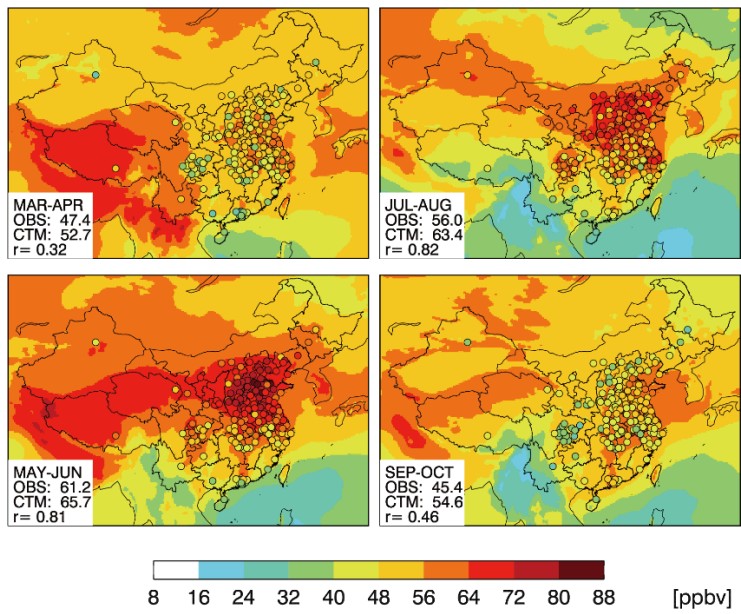

**Figure 2.** Spatial distribution of observed (circles) and simulated (shaded) bimonthly mean surface daily maximum 8-h average (MDA8) ozone concentrations over China averaged for 2016-2017. Observations at the 169 major cities are plotted over the GEOS-Chem model results. Observed (OBS) and simulated (CTM) bimonthly mean values averaged for the 169 cities and their correlation coefficients (*r*) are shown inset.





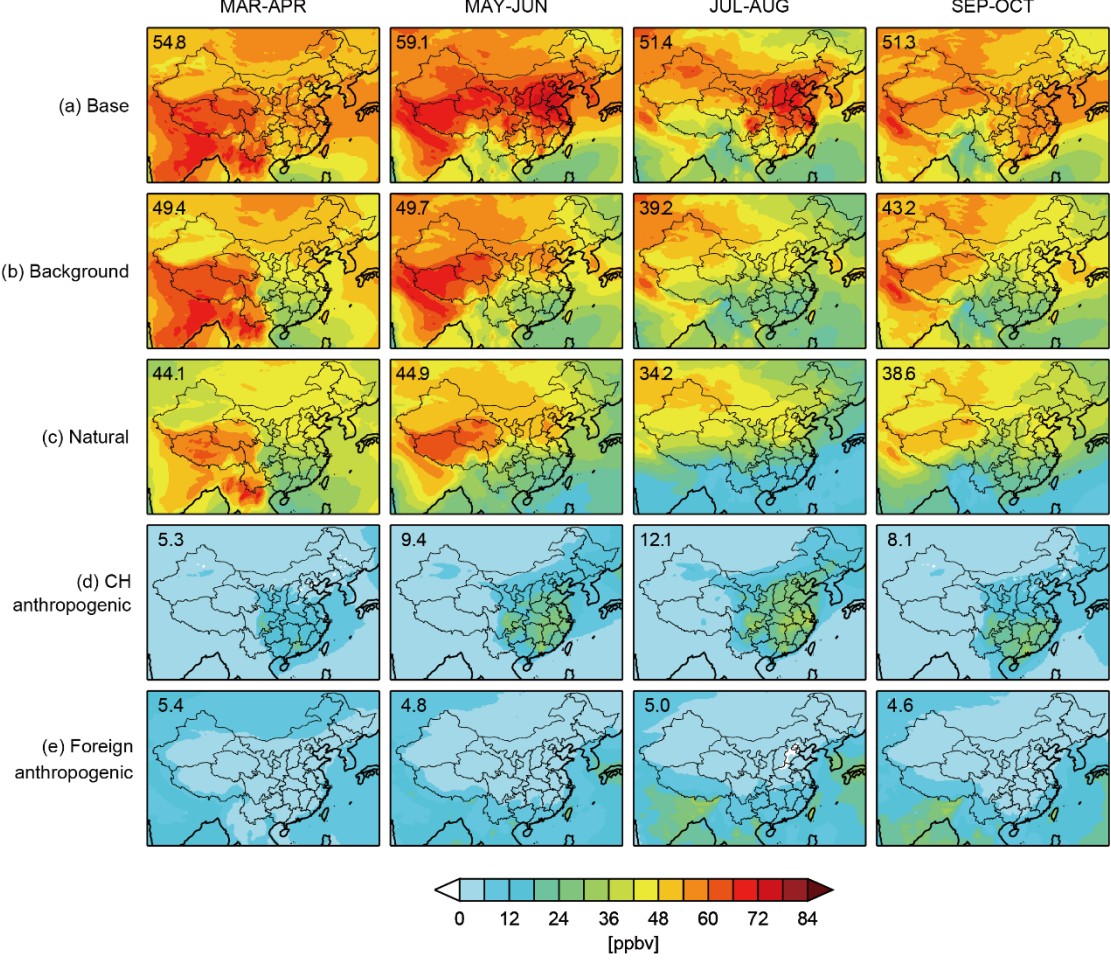

**Figure 3.** Simulated bimonthly mean surface MDA8 ozone over China and the    source attributions averaged for 2016-2017. Plots are (a) surface MDA8 ozone from BASE simulation, (b) Chinese background MDA8 ozone estimated from the noCH simulation in which Chinese anthropogenic emissions are turned off in the model, (c) natural MDA8 ozone estimated from the noGLOBAL simulation in which global anthropogenic emissions are turned off in the model, (d) Chinese anthropogenic MDA8 ozone enhancement diagnosed as the difference between the BASE simulation and the noCH simulation, and (e) foreign anthropogenic MDA8 ozone enhancement diagnosed as the difference between the noCH simulation and the noGLOBAL simulation. The mean values averaged over China are shown inset.





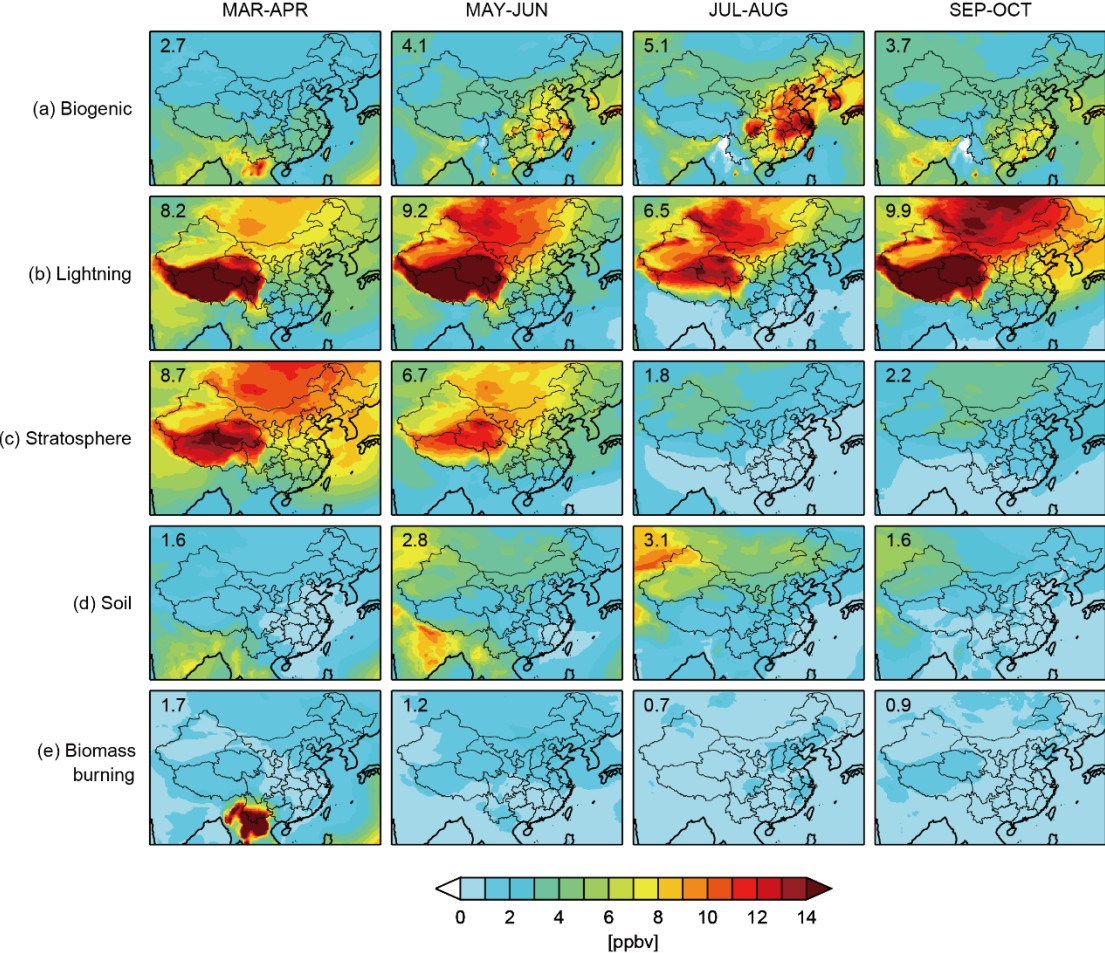

**Figure 4.** Same as Figure 3, but for ozone enhancements from (a) biogenic volatile organic compounds emissions, (b) lightning $NO_x$ emissions, (c) the stratosphere, (d) soil $NO_x$ emissions, and (e) biomass burning emissions. The sources attributions are diagnosed as the difference between the BASE simulation and a simulation with the specific source turned off, expect for ozone from the stratosphere which is diagnosed by the TagO3 simulation as described in the text.





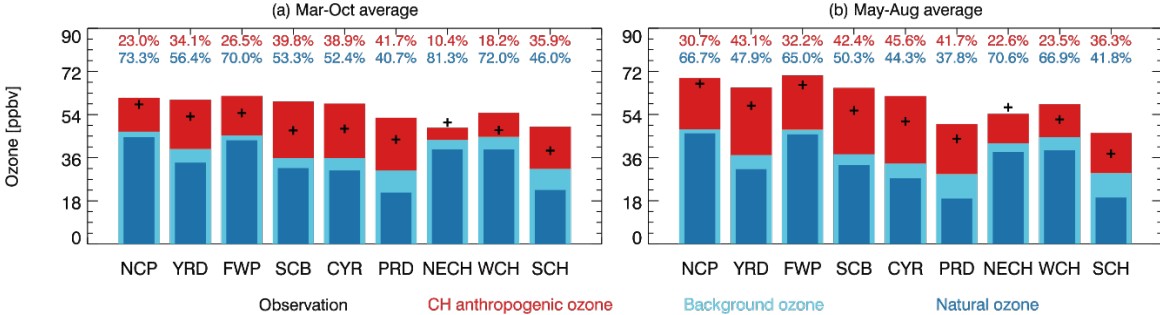

**Figure 5.** Simulated MDA8 ozone and the source attributions at nine city clusters in 2016-2017 averaged for (a) March to October and (b) May to August. The black pluses denote the observed mean ozone MDA8 values. The bars present simulated mean ozone MDA8 values, with the domestic anthropogenic ozone contribution in red, the background ozone contribution in light blue, and the natural sources in dark blue.

The percentage of domestic anthropogenic contribution (red) and natural contribution (dark blue) for each cluster are shown inset.







**Figure 6.** Simulated daily MDA8 ozone (x-axis) and corresponding background contributions (y-axis) at the nine city clusters for March-October 2017 with different months denoted in different colors. Also shown are box-and-whisker plots (minimum, 25th, 50th, and 75th percentiles, and maximum) of domestic anthropogenic ozone enhancements (y-axis) over 10-ppbv bins of simulated ozone concentrations. The 980 1:1 line and number of cities in each cluster are shown inset.





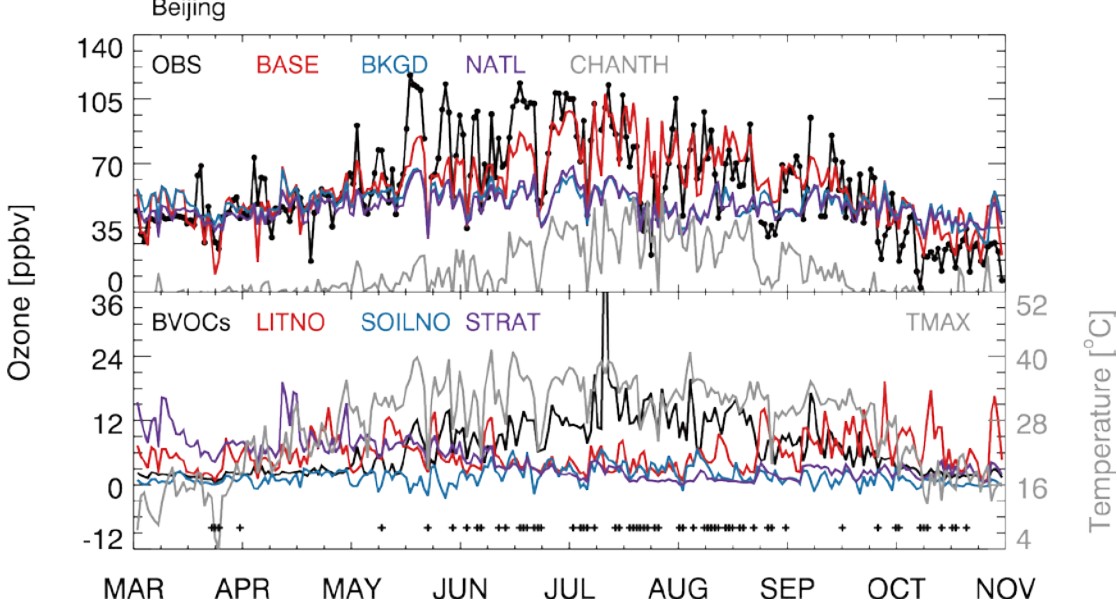

**Figure 7.** Time series of observed and simulated daily MDA8 ozone concentrations, and source attributions at Beijing in March-October 2017. The top panel shows observed MDA8 ozone (OBS, black), simulated ozone (BASE, red), and estimated background ozone (BKGD, blue), natural ozone (NATL, purple), and domestic anthropogenic ozone (CHANTH, grey). The bottom panel shows the ozone enhancement from biogenic VOCs (BVOCs, black), lightning $NO_x$ (LITNO, red), soil $NO_x$ (SOILNO, blue), and the stratosphere (STRAT, purple). Estimation of ozone sources attributions are described in the text. Also shown are daily maximum 2-meter air temperature (TMAX, grey, right y-axis) from the GEOS-FP data. The black pluses denote days with precipitation recorded from GEOS-FP data.

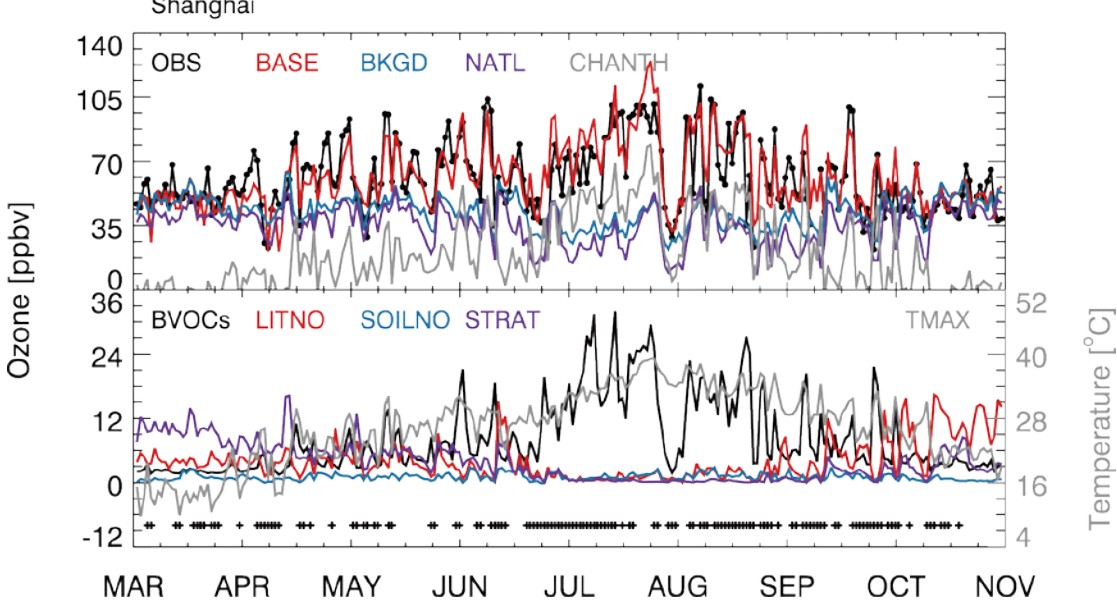

**Figure 8.** Same as Figure 7 but for Shanghai city.

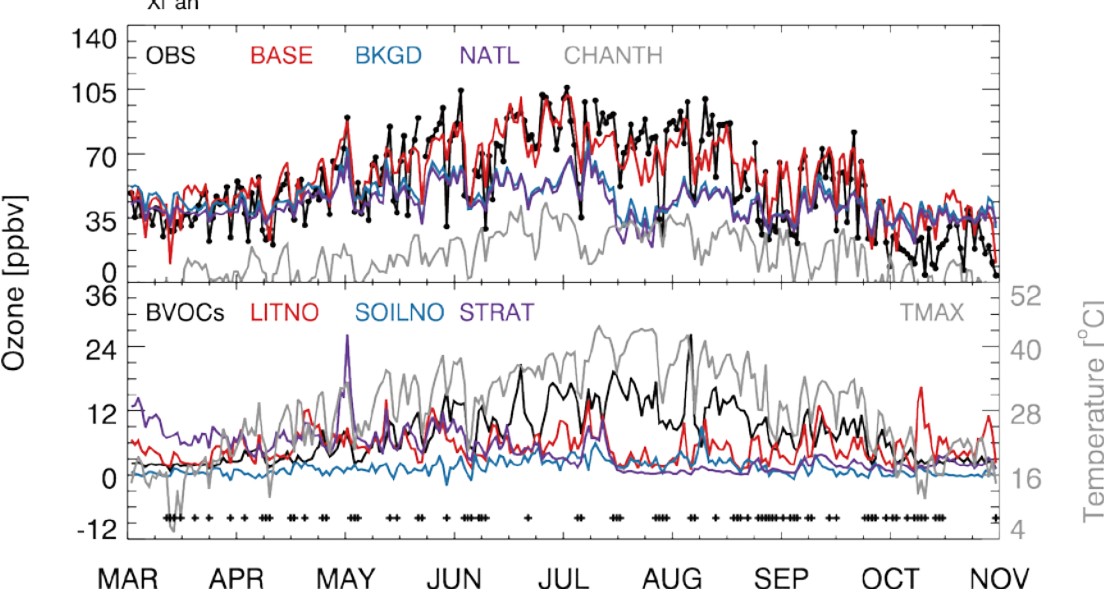

**Figure 9.** Same as Figure 7 but for Xi'an city.



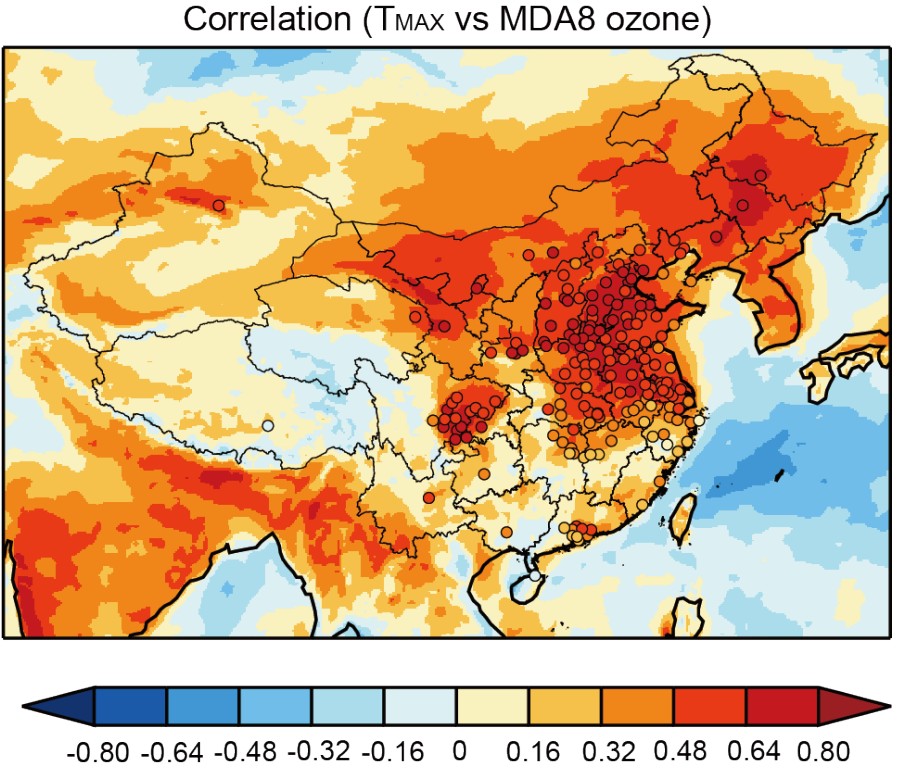

**Figure 10.** Observed (circles) and simulated (shaded) correlation coefficients between deseasonalized surface daily MDA8 values and maximum 2-meter air temperature (TMAX) for May- August 2016-2017.




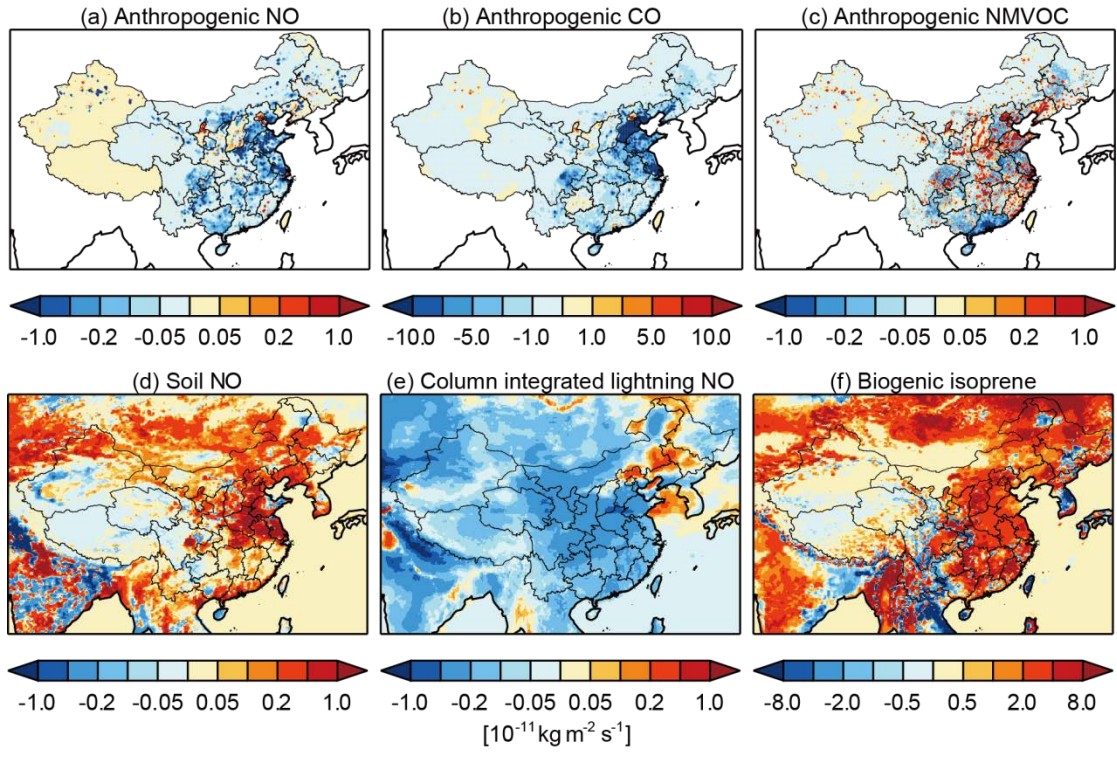

**Figure 11.** Differences in May-August mean (a) anthropogenic NO emissions, (b) anthropogenic CO emissions, (c) anthropogenic NMVOC emissions, (d) soil NO emissions, (e) column integrated lightning NO emissions, and (f) biogenic isoprene emissions between 2017 and 2016 (2017 minus 2016 conditions).




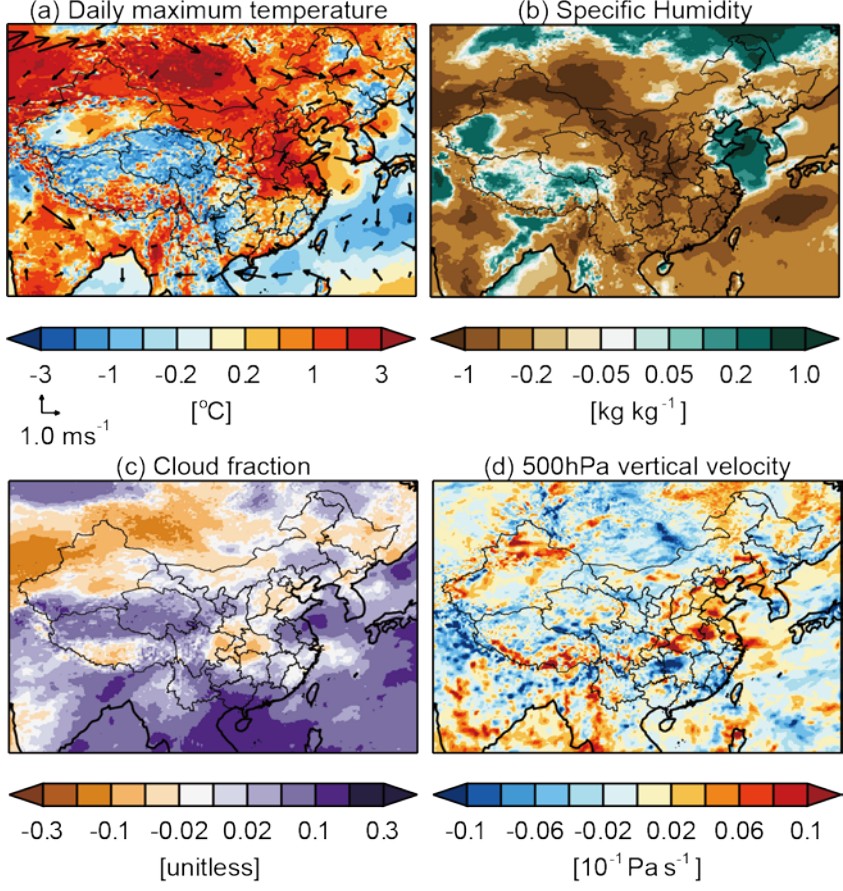

**Figure 12.** Differences in May-August mean (a) daily maximum 2-m air temperature with 850hPa wind vectors over-plotted, (b) specific humidity, (c) cloud cover faction, and (d) 500 hPa vertical velocity between 2017 and 2016.

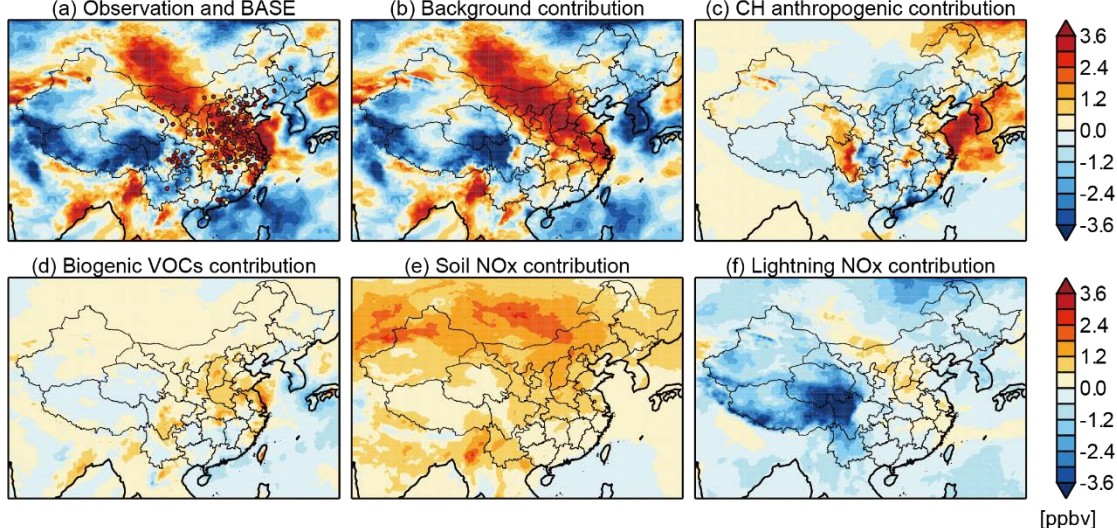

**Figure 13.** Differences in May-August mean (a) observed (circles) and simulated ozone (shaded) surface MDA8 ozone concentration, (b) background ozone, (c) domestic anthropogenic ozone contribution, (d) biogenic VOCs ozone contribution, (e) soil $NO_x$ ozone contribution, and (f) lightning $NO_x$ ozone enhancement between 2017 and 2016.

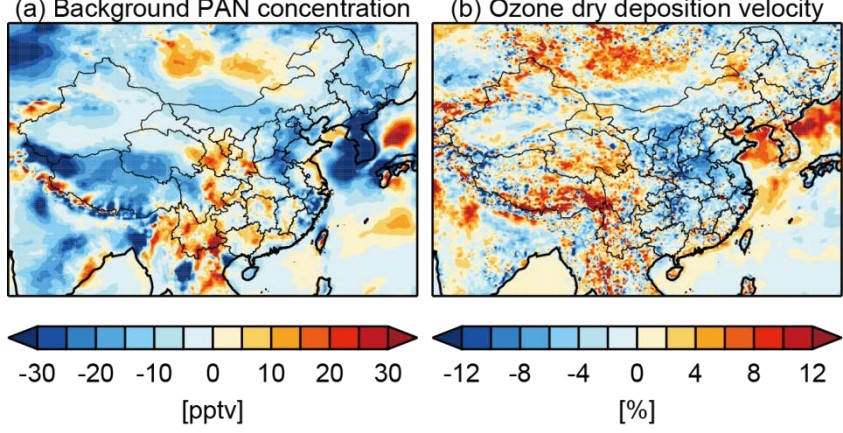

**Figure 14.** Differences in May-August mean (a) background PAN concentration and (b) ozone dry deposition velocity between 2017 and 2016.



**Table 1.** List of the 169 "key cities" for air pollution monitoring in China categorized by different regions[1]

| Region | Province | City |
|---|---|---|
| NCP (55) | Beijing (1) | Beijing |
| | Tianjin (1) | Tianjin |
| | Hebei (11) | Shijiazhuang, Tangshan, Qinhuangdao, Handan; Xingtai, Baoding, Zhangjiakou, Chengde, Cangzhou, Langfang, Hengshui |
| | Shanxi (7) | Taiyuan, Datong, Shuozhou, Xinzhou, Yangquan, Changzhi, Jincheng |
| | Shandong (15) | Jinan, Qingdao, Zibo, Zaozhuang, Dongying, Weifang, Jining, Taian, Rizhao, Laiwu, Linyi, Dezhou, Liaocheng, Binzhou, Heze |
| | Henan (15) | Zhengzhou, Kaifeng, Pingdingshan, Anyang, Hebi, Xinxiang, Jiaozuo, Puyang, Xuchang, Luohe, Nanyang, Shangqiu, Xinyang, Zhoukou, Zhumadian |
| | Inner Mongolia (2) | Huhehaote, Baotou |
| | Liaoning (3) | Chaoyang, Jinzhou, Huludao |
| YRD (41) | Shanghai (1) | Shanghai |
| | Jiangsu (13) | Nanjing, Wuxi, Xuzhou, Changzhou, Suzhou, Nantong, Lianyungang, Huaian, Yancheng, Yangzhou, Zhenjiang, Taizhou, Suqian |
| | Zhejiang (11) | Hangzhou, Ningbo, Wenzhou, Shaoxing, Huzhou, Jiaxing, Jinhua, Quzhou, Taizhou, Lishui, Zhoushan |
| | Anhui (16) | Hefei, Wuhu, Bengbu, Huaian, Maanshan, Huaibei, Tongling, Anqing, Huangshan, Fuyang, Suzhou, Chuzhou, Lu'an, Xuancheng, Chizhou, Bozhou |
| FWP (11) | Shanxi (4) | Lvliang, Jinzhong, Linfen, Yuncheng |
| | Henan (2) | Luoyang, Sanmenxia |
| | Shaanxi (5) | Xi'an, Xianyang, Baoji, Tongchuan, Weinan |
| SCB (16) | Chongqing (1) | Chongqing |
| | Sichuan (15) | Chengdu, Zigong, Luzhou, Deyang, Mianyang, Suining, Neijiang, Leshan, Meishan, Yibin, Ya'an, Ziyang, Nanchong, Guangan, Dazhou |
| CYR | Hubei (11) | Wuhan, Xianning, Xiaogan, Huanggang, Huangshi, |




| (22) | | Ezhou, Xiangyang, Yichang, Jingmen, Jingzhou, Suizhou |
|---|---|---|
| | Jiangxi (5) | Nanchang, Pingxiang, Xinyu, Yichun, Jiujiang |
| | Hunan (6) | Changsh, Zhuzhou, Xiangtan, Yueyang, Changde, Yiyang |
| PRD (9) | Guangdong (9) | Guangzhou, Shenzhen, Foshan, Zhuhai, Jiangmen, Zhaoqing, Huizhou, Dongguan, Zhongshan |
| NECH (4) | Liaoning (2) | Shenyang, Dalian |
| | Jilin (1) | Changchun |
| | Heilongjiang (1) | Dalian |
| WCH (7) | Xinjiang (1) | Urumqi |
| | Qinghai (1) | Xining |
| | Ningxia (1) | Yinchuan |
| | Gansu (1) | Lanzhou |
| | Tibet (1) | Lasa |
| | Guizhou (1) | Guiyang |
| | Yunnan (1) | Kunming |
| SCH (4) | Fujian (2) | Fuzhou, Xiamen |
| | Guangxi (1) | Nanning |
| | Hainan (1) | Haikou |

[1]This city list is updated in July 2018 and can be accessed from http://www.mee.gov.cn/hjzl/dqhj/cskqzlzkyb/201809/P020180905326235405574.pdf (in Chinese). Locations of the cites are shown in Figure 1. Abbreviations are NCP: North China Plain, YRD: Yangtze River Delta, FWP: Fenwei Plain, SCB: Sichuan Basin, CYR: Central Yangtze River Plain, PRD: Pearl River Delta, NECH: northeastern China, WCH: western China, SCH: southern China. Numbers of cities for each region/province are shown in parenthesis.



**Table 2.** Bimonthly mean anthropogenic and natural sources over China used in the model.

|  |  | MAR-APR | MAY-JUN | JUL-AUG | SEP-OCT |
|---|---|---|---|---|---|
| Anthropogenic emissions (Tg) | | | | | |
| NO | 2016 | 2.45 | 2.41 | 2.43 | 2.44 |
|  | 2017 | 2.39 | 2.35 | 2.37 | 2.39 |
| CO | 2016 | 23.17 | 19.58 | 19.39 | 20.01 |
|  | 2017 | 22.25 | 18.86 | 18.67 | 19.30 |
| NMVOC | 2016 | 4.67 | 4.66 | 4.48 | 4.68 |
|  | 2017 | 4.71 | 4.73 | 4.53 | 4.74 |
| Natural emissions (Tg) | | | | | |
| Soil NO | 2016 | 0.08 | 0.19 | 0.20 | 0.10 |
|  | 2017 | 0.08 | 0.24 | 0.25 | 0.11 |
| Lightning NO | 2016 | 0.04 | 0.12 | 0.20 | 0.03 |
|  | 2017 | 0.01 | 0.06 | 0.15 | 0.02 |
| Biogenic isoprene | 2016 | 1.91 | 7.11 | 10.53 | 3.60 |
|  | 2017 | 1.84 | 7.50 | 11.50 | 3.82 |
| Biomass burning CO | 2014 | 2.12 | 0.62 | 0.72 | 0.78 |

1050





**Table 3.** Configurations of GEOS-Chem simulations in this study.

| Simulation | Description |
| --- | --- |
| BASE | Full-chemistry with year-specific anthropogenic and natural emissions as described in the text |
| noGLOBE | Same as BASE but without global anthropogenic emissions. Ozone concentrations from this simulation are defined as natural background ozone (natural ozone). |
| noCH | Same as BASE but without domestic anthropogenic emissions over China. Ozone concentrations from this simulation are defined as Chinese background ozone (background ozone). |
| noSOIL | Same as BASE but without soil emissions. |
| noBVOC | Same as BASE but without biogenic VOCs emissions. |
| noLIGHT | Same as BASE but without lightning emissions. |
| noBB | Same as BASE but without biomass burning emissions. |
| TagO3 | A simulation labels stratospheric ozone (ozone produced in the stratosphere from photolysis of molecular oxygen) as a tagged tracer. The simulation is driven by ozone production rates and loss frequencies archived from the BASE simulation. |

1055