# Peer review of "Exploring 2016-2017 surface ozone pollution over China: source contributions and meteorological influences"

_Atmospheric Chemistry and Physics, 2019_

## Referee Comment (RC1) · Anonymous Referee #1 · 5 Apr 2019

The authors examined source contributions of anthropogenic, background, and individual natural sources to surface ozone over China, as well as their differences between 2016 and 2017, in a chemical transport model GEOS-Chem. They found that natural background accounted 70-80% of surface ozone in China. Domestic anthropogenic sources contributed 30% during May-August and up to 69% during polluted ozone days. Ozone increases in 2017 relative to 2016 are due to hotter and dryer weather conditions. This topic is of interest and the manuscript is well written. I would suggest publishing after addressing my comments below.

My main concern is about the method for calculating source contribution. The authors considered the contribution from one source as the differences in ozone between BASE and sensitivity simulations with individual source emissions turned off. They did not consider the nonlinearity of source contribution. As the authors mentioned in the discussion that zero-out and 20% off emission would produce different contribution value. Ozone chemistry is complex. The nonlinearity exists non only in the amount of emission perturbed, but also in the species of emissions (VOCs or NOx) and the location of emission perturbed. Directly comparing the source contribution derived from the differences of two simulation could be biased. The authors needs to quantify how large the nonlinearity would perturb the results with more simulations or, at least, discuss the potential biases of results related to the nonlinearity.

Minor comments:

The ozone contribution from each source was estimated as the ozone difference between the BASE simulation and each sensitivity simulation for BVOC, lightning NOx, soil NOx and biomass burning emissions. But the stratospheric contribution was quantified using an ozone tagging method. Is it appropriate to compare contributions derived from two different methods? How large the uncertainty it has?

How does the model treat the emission injection height? In the recent study, Yang et al. (2019) found that uncertainty in anthropogenic emission height strongly affects surface SO2 concentration by about 80%. The inaccurate emission injection height can also lead to the bias of simulated surface ozone concentration.

Line 320: What are the 'interactional' and 'pure' contributions?

Table: I suggest adding values for percentage change.

Reference:
Yang Y., Smith S. J., Wang H., Lou S., and Rasch P. J., Impact of anthropogenic emission injection height uncertainty on global sulfur dioxide and aerosol distribution, J. Geophys. Res. Atmos., accepted, 2019.

---

## Referee Comment (RC2) · Anonymous Referee #2 · 27 Apr 2019

The manuscript represents a comprehensive modeling analysis of background ozone and its origins in China for a recent period of 2016 and 2017 and explains how differences in meteorology between the two years led to ozone differences. It uses a high-resolution nested-grid version of the global chemical transport model (GEOS-Chem) and does a good job comparing to previously published coarser-resolution GEOS-Chem modeling studies of background ozone in China. The methodology used to quantify background ozone follows the conventional emission zeroing-out approach used by the literature. The manuscript is well-written and well organized. I recommend publication after the following comments are addressed.

[Figure]

Figure 3: I was surprised that lightning NOx has the largest contribution to background ozone in western China exceeding stratospheric ozone and that this contribution does not have a clear seasonality. I would expect lightning NOx to peak in summer; in fact, the figure shows it is lowest in summer (6.5 ppbv vs. 8-9 ppbv in other seasons). Is there a way to validate this result with observations or by comparing with literature values (if any). It will be helpful to put an uncertainty estimate to these numbers. From the model validation plots in Figure 2, I can see the model overestimates surface ozone in western China although there is just a couple of sites available. Could the overestimation be partly caused by an overestimation of ozone contributed by lightning NOx?

Pg 10, line 292-295: This statement needs to be elaborated; otherwise it sounds superficial. What are the possible interactions between domestic and foreign anthropogenic emissions and to what direction would these interactions affect ozone (i.e. increase or decrease)?

Pg 13, line 380-385: The statement that "the missing rest can be largely explained by contributions from global methane" is too assertive. Other factors such as the interactions between different sources may also play a role which will not be captured by the sensitivity simulations by zeroing off individual emissions.

―――――――――――――――――――

---

## Author Comment (AC1) · 31 May 2019

**Reviewer #1**

**Comment#1-1:** The authors examined source contributions of anthropogenic, background, and individual natural sources to surface ozone over China, as well as their differences between 2016 and 2017, in a chemical transport model GEOS-Chem. They found that natural background accounted 70-80% of surface ozone in China. Domestic anthropogenic sources contributed 30% during May-August and up to 69% during polluted ozone days. Ozone increases in 2017 relative to 2016 are due to hotter and dryer weather conditions. This topic is of interest and the manuscript is well written. I would suggest publishing after addressing my comments below.

**Response#1-1: We thank the reviewer for the valuable comments. All of them have been implemented in the revised manuscript. Please see our itemized responses below.**

**Comment#1-2:** My main concern is about the method for calculating source contribution. The authors considered the contribution from one source as the differences in ozone between BASE and sensitivity simulations with individual source emissions turned off. They did not consider the nonlinearity of source contribution. As the authors mentioned in the discussion that zero-out and 20% off emission would produce different contribution value. Ozone chemistry is complex. The nonlinearity exists non only in the amount of emission perturbed, but also in the species of emissions (VOCs or NOx) and the location of emission perturbed. Directly comparing the source contribution derived from the differences of two simulation could be biased. The authors needs to quantify how large the nonlinearity would perturb the results with more simulations or, at least, discuss the potential biases of results related to the nonlinearity.

**Response#1-2: To addressed the concern, we have further conducted an additional high-resolution simulation for July 2017 by decreasing Chinese anthropogenic emissions by 20% in the model, and presented the results in Figure S9. We state in the Section 5 (Discussions and Conclusion) "To further estimate the non-linear response of ozone to changes of sources, we conduct an additional sensitivity simulation with Chinese domestic anthropogenic emissions reduced by 20% in the model for July 2017. As shown in Figure S9, compared to the zero-out method, the 20% perturbation method estimates much lower domestic anthropogenic contributions (6.8 ppbv vs. 11.8 ppbv averaged over China in July 2017) with similar spatial distributions, which is consistent with Ni et al. (2018). Such strong non-linear responses of ozone to precursor emissions in China reflects that more stringent anthropogenic emission control measures are required to mitigate ozone pollutions.".**

[Figure]

**Figure S9.** Spatial distributions of Chinese domestic anthropogenic contributions to surface MDA8

ozone in 2017 estimated from (a) zero-out methods (difference between the BASE simulation and the noCH simulation), (b) 20% perturbation method (five times of difference between the BASE simulation and the CH20off simulation). (c) shows the difference between (a) and (b).

**Comment#1-3:** The ozone contribution from each source was estimated as the ozone difference between the BASE simulation and each sensitivity simulation for BVOC, lightning NOx, soil NOx and biomass burning emissions. But the stratospheric contribution was quantified using an ozone tagging method. Is it appropriate to compare contributions derived from two different methods? How large the uncertainty it has?

**Response#1-3: Thanks for pointing it out. Ozone in the lower stratosphere has much a longer lifetime (several years). Using the zero-out method to estimate stratospheric influences on tropospheric ozone in the model requires a very long spin-up time (more than ten years) and it is computationally unaffordable. Therefore, the tagged ozone simulation is typically used to quantify stratospheric influences on tropospheric ozone in chemical transport models. We now explain in the text: "Due to the long lifetime of ozone in the lower stratosphere (~years, Wang et al., 1998), the stratospheric contribution to tropospheric ozone is typically quantified using the tagged ozone simulation (TagO3) (Wang et al., 1998; Zhang et al., 2014), instead of perturbing the stratospheric ozone chemistry that requires a long spin-up time."**

**Reference in the text:**

Wang, Y., Jacob, D. J., and Logan, J. A.: Global simulation of tropospheric O3-NOx-hydrocarbon chemistry: 3. Origin of tropospheric ozone and effects of nonmethane hydrocarbons, J. Geophys. Res., 103, 10757-10767, http://doi.org/10.1029/98jd00156, 1998.

**Comment#1-4:** How does the model treat the emission injection height? In the recent study, Yang et al. (2019) found that uncertainty in anthropogenic emission height strongly affects surface SO2 concentration by about 80%. The inaccurate emission injection height can also lead to the bias of simulated surface ozone concentration.

**Response#1-4: Thanks for pointing it out. We now state in the Section 2.2: "All anthropogenic emissions are emitted at the lowest layer in the model with a thickness of 120 m. A recent study found that uncertainties in industrial emission injection height could affect surface $SO_2$ concentrations by about 80% (Yang et al., 2019), and this may further affect surface ozone concentrations that requires further study."**

**Reference in text:**

Yang, Y., Smith, S. J., Wang, H., Lou, S., and Rasch, P. J.: Impact of Anthropogenic Emission Injection Height Uncertainty on Global Sulfur Dioxide and Aerosol Distribution, J. Geophys. Res., http://doi.org/10.1029/2018jd030001, 2019.

**Comment#1-5:** Line 320: What are the 'interactional' and 'pure' contributions?

**Response#1-5: We now state in the text to further elaborate the 'interactional' and 'pure' contributions: "These values are considered as the actual contributions of BVOCs on ozone, which include the pure contributions, i.e., contributions of BVOCs on ozone in the absence of**

**all other sources, and the interactional effects of BVOCs with other sources.”**

**Comment#1-6:** Table: I suggest adding values for percentage change.
**Response#1-6: Changed as suggested.**

---

## Author Comment (AC2) · 31 May 2019

**Reviewer #2:**

**Comment#2-1:** The manuscript represents a comprehensive modeling analysis of background ozone and its origins in China for a recent period of 2016 and 2017 and explains how differences in meteorology between the two years led to ozone differences. It uses a high resolution nested-grid version of the global chemical transport model (GEOS-Chem) and does a good job comparing to previously published coarser-resolution GEOS-Chem modeling studies of background ozone in China. The methodology used to quantify background ozone follows the conventional emission zeroing-out approach used by the literature. The manuscript is well-written and well organized. I recommend publication after the following comments are addressed.

**Response#2-1: We thank the reviewer for the valuable comments. All of them have been implemented in the revised manuscript. Please see our itemized responses below.**

**Comment#2-2:** Figure 3: I was surprised that lightning NOx has the largest contribution to background ozone in western China exceeding stratospheric ozone and that this contribution does not have a clear seasonality. I would expect lightning NOx to peak in summer; in fact, the figure shows it is lowest in summer (6.5 ppbv vs. 8-9 ppbv in other seasons). Is there a way to validate this result with observations or by comparing with literature values (if any). It will be helpful to put an uncertainty estimate to these numbers. From the model validation plots in Figure 2, I can see the model overestimates surface ozone in western China although there is just a couple of sites available. Could the overestimation be partly caused by an overestimation of ozone contributed by lightning NOx?

**Response#2-2: Thanks for pointing it out. The magnitude and spatial patterns of lightning ozone enhancements in our simulation are generally comparable with Murray et al. (2016), although they did not provide seasonal values. We find although higher lightning NOx emissions in summer (Table 3 and Fig. S5), the lightning influences on surface ozone are also affected by elevation and vertical transport. The following text has been added. We have also added the standard deviations for all source attributions in Figures 3 and 4 following the reviewer's suggestions.**

**We now state in the text: "Lightning NO$_x$ emissions increase bimonthly mean surface MDA8 ozone by 6.5-9.9 ppbv averaged over China, with the largest contributions (typically more than 12 ppbv) found over the Tibetan Plateau (Fig. 4b). The large lightning ozone enhancements over the western China (annual mean > 7 ppbv) were also simulated by Murray et al. (2016), and these values were higher than those over the western US (annual mean of 3-5 ppbv). However, the model may overestimate springtime lightning ozone enhancements over the Tibetan Plateau as the model shows high surface ozone biases of more than 15 ppbv in spring over this region (Fig.2)." and "Although lightning NO$_x$ emissions are larger in July-August, the shorter ozone lifetime and stronger upward transport over central eastern China in these months can suppress downward mixing of lightning ozone enhancements to the surface (Fig. S4), resulting in their minimum influences there in the period. This also partly explains the decreases of surface ozone from spring to summer in the western China as shown in Fig. 2."**

**Reference in the text:**

Murray, L.: Lightning NO$_x$ and Impacts on Air Quality, Current Pollution Reports, 2, 115–133, https://doi.org/10.1007/s40726-016-0031-7, 2016.

**Comment#2-3:** Pg 10, line 292-295: This statement needs to be elaborated; otherwise it sounds superficial. What are the possible interactions between domestic and foreign anthropogenic emissions and to what direction would these interactions affect ozone (i.e. increase or decrease)?

**Response#2-3: Thanks for pointing it out. We now rewrite the statement: "The foreign anthropogenic contributions we estimate here (~5 ppbv for all seasons averaged over China), however, may underestimate their true contribution, since they are derived in the absence of Chinese domestic anthropogenic emissions and thus do not consider possible interactions with domestic emissions, e.g., ozone produced by foreign precursor enhancements reacted with domestic anthropogenic emissions."**

**Comment#2-4:** Pg 13, line 380-385: The statement that "the missing rest can be largely explained by contributions from global methane" is too assertive. Other factors such as the interactions between different sources may also play a role which will not be captured by the sensitivity simulations by zeroing off individual emissions.

**Response#2-4: We now state in the text: "The missing rest can be largely attributed by contributions from global methane." and "The discrepancies may be also due to non-linear interactional effects between different sources that are not captured by the sensitivity simulations with individual emissions turned off."**